

# Network-risk: an open GIS toolbox for estimating the implications of transportation network damage due to natural hazards, tested for Bucharest, Romania

Dragos Toma-Danila[1], Iuliana Armas[2], Alexandru Tiganescu[1]

[1]National Institute for Earth Physics, Magurele, Ilfov, 077125, Romania
[2]Faculty of Geography, University of Bucharest, Bucharest, 010041, Romania

*Correspondence to*: Dragos Toma-Danila (toma@infp.ro)

**Abstract.** Due to their widespread and continuous expansion, transportation networks are considerably exposed to natural hazards such as earthquakes, floods, landslides or hurricanes. The vulnerability of specific segments and structures such as

bridges, tunnels, pumps or storage tanks can translate not only in direct losses but also in significant indirect losses at systemic level. Cascading effects such as post-event traffic congestion, building debris or tsunamis can contribute to an even greater level of risk. To support the effort of modelling the natural hazards implications at full transportation network scale, we developed a new applicable framework relying on i) GIS to define, geo-spatially analyze and represent transportation networks; ii) methods for determining the probability of network segments to fail due to natural hazard effects; iii) Monte Carlo

simulation for multiple scenario generation; iv) methods (using Dijkstra algorithm) to analyze the implications of connectivity loss on emergency intervention times and transit disruption, v) correlations with other vulnerability and risk indicators. Currently, the framework is integrated in ArcGIS Desktop as a toolbox (entitled "Network-risk") - making use of the Model Builder functions and being free for download and customize. Network-risk is an attempt to bring together interdisciplinary research with the goal of creating an automated solution to deliver insights on how a transportation network can be affected

by natural hazards, directly and indirectly, aiding in risk evaluation and mitigation planning. In this article we present and test Network-risk at full urban scale, for the entire road network of Bucharest - one of the world's most exposed capitals due to earthquakes, with high seismic hazard values and a vulnerable building stock, but also significant traffic congestion problems not yet quantified in risk analyses.

## 1 Introduction

The complexity, size and exposure of our society to natural hazards has significantly increased in the last decades (Fleischhauer, 2008), and will keep on doing so. Transportation networks are one of the fundamental pillars of development and support for countries and regions, whether they are represented by road, railway, pipeline, communications and other types of networks. Also, we should start to see them in our analysis not just as isolated systems, but as interconnected infrastructures. Transportation networks are a requirement for almost every inhabited place - residential, commercial or industrial, and they



continue to upgrade per location and also expand. As such, they become more and more exposed, if not also more vulnerable. Recent large-scale natural hazard events, such as the earthquakes (in Italy 2016 and 2009, Nepal 2015, Haiti 2010, China 2008 etc.) accompanied in some cases by very destructive tsunamis (Japan 2011 or Indonesia 2018 and 2004), hurricanes and typhoons (in Mozambique 2019, Puerto Rico 2017, Philippines 2013 and 2012, Myanmar 2008 or USA 2005) or heat-waves (constant in the last years in countries such as USA, Australia, Greece or Spain), proved that transportation networks are

extremely vulnerable but also vital immediately after the event occurrence. Directly contributing on the economic loss balance of such events, transportation networks have a more and more significant percentage, especially in developed countries (Wilkerson, 2016), and when adding also the indirect losses (hard to quantify), it's even more obvious that their vulnerability needs to be reduced.

Transportation networks are very important both immediately after a hazardous event - constituting support for emergency
intervention, and long time after - in the recovery and prevention phases. Not to mention also that their damage can inflict direct risks - collapse of vehicles or trains, fire outbreaks etc. Functionality and redundancy are essentials, in order to ensure the reduction of socio-economic losses. In the post-disaster reaction phase, road networks are very important, since they link almost all destinations, but in some cases other transportation networks can be more relevant: railways, maritime or aerial networks. Communication networks are also critical in all disaster cycle phases. Utilities are important for a faster recovery

and overall, for ensuring resilience. Previous experiences show that transportation networks can be mostly affected:

- Directly: by the collapse of critical components such as bridges, tunnels, storage tanks, pumps etc., cracks in roadways due to ground motion effects (settlement, liquefaction), railway displacement, pipe cracks;
- Indirectly: road blockage due to collapsed buildings (especially in urban areas), blockage due to triggered landslides, flooding or tsunamis, blockage due to traffic congestion generated by post-disaster behavior etc.

For studying the impact of natural disasters on transportation networks, multi and inter-disciplinary approaches are needed, combining methods belonging to geosciences, engineering, sociology, economy or informatics. Also, multiple perspectives need to be considered (Franchin et al., 2011):

- Temporal (the disaster management cycle)
- Spatial: local (structural element studies), regional, national or multinational
- Actor involved (Level of management)

In the last two decades, significant progress has been achieved in transportation network vulnerability and risk analysis - not just at structural level but also at functional level. For a comprehensive review we recommend the studies of Franchin et al. (2011) in the framework of the Syner-G Project, by Miller (2014), Tesfamariam & Goda (2013) or Kiremidjian et al. (2007). These reveal that the fundamental steps in evaluating the seismic risk of transportation networks are:

- the proper definition of the network, with detailed knowledge at the level of component characteristics and connectivity. One of the problems is still the lack of official data: in developed countries there can be used good and updated GIS databases, however in most other countries transportation network data (at least for roads or railways) is not well officially defined, therefore alternative sources need to be used, such as OpenStreetMap (open-source) or commercial products



(Google Maps, Here Maps etc.). There are currently many software solutions providing network definition (including
AutoCAD Civil 3D, OpenRoads or ESurvey Road Network), but not so many providing risk analysis capabilities; among
them are ArcGIS for Desktop with the Network Analyst extension, PTV Visum/Vissim, Maeviz/Eqvis or STREET;

-  the determination of direct damage probability of individual components. For this, earthquake engineering analysis
methods are mostly used, such as dynamical elastic and inelastic analysis using grids and numerical methods: finite
element method, pushover or time-history analysis, response spectra etc. A good synthesis of these methods can be found
in Costa (2003) and Crowley et al. (2011);

-  the need to define relevant performance indicators, reflecting time or cost differences between pre and post-disaster
network behavior; many performance indicators for networks can be found in literature, some of the most common at
system level being Driver Delay, Simple/Weighted Connectivity Loss (Poljanšek et al., 2011; Pinto et al., 2012), System
Serviceability Index (Wang et al, 2010) or Serviceability Ratio (Adachi and Ellingwood, 2008).

In the even more recent years, new technologies such as Internet of Things devices, Big Data, Remote Sensing, drone
surveillance, low-cost sensors and Machine Learning started to be quickly adopted since they can provide practical solutions
for transportation network data collection and analysis. It is expected that the impact of future natural hazards on transportation
networks will be much better recorded (as shown by Voumard et al., 2018), allowing for a much needed validation of risk
models and opportunities to better understand what went wrong.

In order to analyze systemic risk (and not just component risk), networks need to be evaluated from the perspective of direct
damage implication on connectivity, traffic changes or new traffic flows created, leading to indirect damage. Recent studies
have addressed these aspects (Franchin et al., 2006, Douglas et al., 2007, Bono & Gutierrez, 2011; Caiado et al., 2012), going
beyond the simple summarization of direct effects and eventually of reconstruction costs generated. These studies also
highlight an important aspect to consider (Pitilakis și Kakderi, 2011): interactions between the components of the system (inter-
interactions) and with components of other systems (intra-interactions).

After analyzing available methodologies and solutions in the field of study, we reached the conclusion that nowadays
capabilities can be better exploited, enabling a more flexible analysis of transportation network implications due to natural
hazards, compared to previous works. We consider that many has been done theoretically and too little practically (at least at
full city scale analysis), and that is what motivated us to create a new GIS solution sharable with the community and applicable
world-wide. In this paper that provides a settled methodology after preliminary studies such as Toma-Danila (2018) or Toma-
Danila et al. (2016), we will focus on:

-  presenting our flexible methodology for evaluating direct and indirect transportation network risk due to natural hazards,
embedded in ArcGIS Desktop as an open-source toolbox - called "Network-risk" (can be downloaded, with user manual
and sample data from www.infp.ro/network-risk);

-  demonstrating its capabilities for a representative case study: implications of earthquakes on the entire urban road network
of Bucharest - one of the most under risk capitals in Europe; results represent an important contribution to emergency
management risk reduction planning.



## 2 Methodology and implementation

The idea behind the proposed methodology is at first sight easy to follow, and it comprise of:

- defining a transportation network in a GIS;

- evaluating which segments could be affected by a natural hazard (directly or indirectly) - accounting also for the damage probability;

- generating random damaged network scenarios based on this probability;

- evaluating which are the implications, in terms of connectivity and serviceability losses and then socio-economic
consequences.

This concept was previously defined in studies such as Hackl et al. (2018), Zanini et al. (2017), Chang et al. (2012) or Argyroudis et al. (2005). However, the way each of the tasks are treated, linked and implemented in GIS is what we consider to be of novelty, allowing among others the consideration of multiple transportation network types (road, railway, utilities etc., represented at local, regional or national level) and of different natural hazards. The methodology, presented in Fig. 1, can
accommodate, for example, the analysis of earthquake implications, where damage will be widespread and building debris and traffic patterns are necessary to consider, as well as a good level of details for network definition; for landslides, the factors to be considered will change, since damage will be much more punctual and random simulations might not be so representative; for flooding, vulnerability analysis of networks such as road or railways will require knowledge on topography - not so representative for earthquake analysis. Still, the methodology will accommodate all these hazard types and influences, as long
as, for example, loss analysis will lead to the identification of affected network segments. There is also flexibility in the way the risk analysis is oriented - toward emergency intervention, economic losses evaluation or urban planning.

Most of the input data (yellow boxes in Fig. 1) is required, also with GIS reference, with the exception that, depending on the analysis type, emergency intervention facilities or origin-destination (OD) pairs will not necessary be needed. In addition, an analysis without typical traffic data can be performed, although representative probably just for night traffic conditions.
The process of building a consistent transportation network, from more or less complex datasets, is an essential part in every network analysis to be performed using Network-risk. To assist in this effort we created a guide, models and layer symbologies for properly converting and editing data, following also the ArcGIS Desktop Network Analyst extension help recommendations. An alternative solution can be to use the ArcGIS OSM editor (https://github.com/Esri/arcgis-osm-editor) for OpenStreetMap data, possible however to have limitations in expressing Z-elevation, or of GRASS GIS v.net. The
converted data is expected by Network-risk to be similar to the sample files provided on the Network-risk webpage, essential columns being "name", "oneway", "F_ZLEV", "T_ZLEV", "hierarchy", "maximum_speed", "FT_minutes" and "TF_minutes" (to which we further columns accounting for traffic, scenario travel times or lack of functionality due to earthquake effects will need to be added).

Both pre and post-earthquake traffic data are highly important, since they show the typical functionality status of the network
and the premises for new traffic jam occurrence, immediately after an earthquake (with correlations also to road segments



blocked by e.g. building debris or bridge collapse), all impacting the risk. Typical traffic data can be retrieved from local data sources (such as traffic management authorities) or from companies taking advantage of new device capabilities, such as Google Traffic, Here Traffic or Waze, providing live (or statistical) data regarding traffic values and reported incidents - although to integrate this data into our framework it will need to be converted as travel speed per road segment, to manipulate
afterwards the network in a custom manner. Other solutions can be to use GPS data from emergency vehicles and the knowledge of their drivers.

The network layer represents the exposure; to evaluate the vulnerability of network segments to a specific natural hazard (or multi-hazard - the analysis can also take this dimension), it is required to associate failure probabilities - using for example vulnerability functions. For individual structures (such as bridges, tunnels, pump facilities, electricity poles) or for buildings
(including network buildings), vulnerability functions are commonly used to determine damage probability or even more: resilience indicators such as closure time or recovery cost. Although it is recommended to use structure-specific (local) functions, considering particular properties of the structure and of the construction practices in the specific country/region, there are currently available fragility function libraries, collected and harmonized in projects such as Hazus or Syner-G, which can be associated, preliminary, to some of the assets in other region. In some cases, analyzing the probability of a building to
collapse can be further linked to the probability of road blockage, due to debris for example (in case of earthquakes, there are equations for this task, such as Moroux et al., 2004, Argyroudis et al., 2005 and Zanini et al., 2017). Also, knowing where affected areas are will contribute to an evaluation of indirect risk - for example, chance of people caught under debris to survive (using results of studies such as Goncharov, 1997 or Coburn and Spence, 2002), given a long intervention reach time.

After including references to the natural hazard, in the form of a map with relevant values to the hazard and vulnerability
functions, the result would be an evaluation of direct possible damage (and implications on the network), and as such a probability of network segment blockage. This will be used for generating random scenario simulations using the Monte Carlo approach (potential acknowledged by Burt and Graham, 1971), in order to test the behavior of the network in multiple probable situations. A probability of 100% for a network segment would indicate certain blockage - very hard to consider for a transportation network, but worst-case scenario could use this value. Also, post-event traffic can be included in these
simulations. Monte Carlo scenarios are usually supposed to come in large number (at least hundreds or thousands of runs), and, depending on the size of the network, the amount of computational time is expected to be considerable. However, in many network analysis cases, the need for a vast number of Monte Carlo scenarios is not really needed to be considerable. The existence of many viable detour alternatives in urban areas (where also network segments are most exposed to damage) or the small number of identified network segments expected to be highly damaged can determine the need of a smaller sample of
Monte Carlo scenarios - that is why the stabilization of aggregated result must be traced.

For estimating post-event traffic patterns, it is need also to include assumptions providing travel speed modifications for road segments located close to affected areas, especially in urban agglomerations. Some hints for determining these patterns can be found in the work of Chang et al. (2012) and Zanini et al. (2017). More complex approaches, relying on individual driver



behavior simulations or decision patterns (as described in Asaithambi & Basheer, 2017 or Munigety & Mathew, 2016) can be implemented.

At the core of the network risk analysis is the Dijkstra algorithm (Sniedovich, 2016), used for computing the shortest distance (in real meters or costs) for various network configurations - pre and post event (for Service Area/Route/Closest Facility/OD Matrix analysis). This algorithm is widely used in systemic network analysis, providing a good balance between precision and performance (Bast et al., 2016). For Service Area analysis - basic for emergency intervention travel time evaluation, we recommended as analysis parameter using Detailed Polygon Generation, with results of prior analysis for identifying inaccessible network areas as barriers, since the results will better reflect small inaccessible areas.

The entire methodology is embedded currently in ArcGIS Desktop Advanced (10.1+ version), with the Network Analyst extension, as the Network-risk toolbox, using ModelBuilder capabilities (Fig. 2) and taking advantage of already available geo-processing algorithms, not yet previously linked however for large-scale network risk analysis in an automated framework, shared with the research community. We chose to split Network-risk in multiple separate modules (such as for network creation, Monte Carlo scenario creation, disrupted network building, service area analysis or aggregation of results into a final index), making it easy to identify errors at different steps. The toolbox is available for download at www.infp.ro/network-risk and is free to use and customize. On the other hand, having the methodology implemented in ArcGIS offers extended analysis support, through cartographic, spatial analyst modules, available basemaps, plug-ins such as ArcCASPER (Shahabi and Wilson, 2014) for computing evacuation routes etc.

Overall, the methodology and Network-risk toolbox can answer to important questions (for emergency management, city planning, commercial, insurance, industrial or real-estate agents and many others), such as:

- How vulnerable is an area due to the direct and indirect implications of natural hazards on the transportation network serving it?
- Which is the socio-economic risk in case of a natural disaster, correlated also with emergency management capabilities?
- Which are the vital access routes in case of a disaster? Are there viable detour routes?
- How would new network segments, hospitals or fire stations contribute to reducing the risk? Where should they be placed?

## 3 Bucharest road network case study, considering seismic hazard

### 3.1 Data and methods

For testing the methodology, we selected one of Europe's most under seismic risk capitals (Toma-Danila and Armas, 2017; Pavel, 2016): Bucharest- a city affected by strong earthquakes in the Vrancea Area (such as the ones on November 10[th], 1940, Mw 7.7 at 150 km depth and on March 4[th], 1977, Mw 7.4 at 94 km depth), waiting for a next major event to happen. Compared to 1977 (when 1578 people died in Romania, from which 90% in Bucharest), the city now faces an additional challenge, beside the high vulnerability of the building stock: the vulnerability of road network. In a city with over 2 million inhabitants in 2018 there are 1.2 million registered vehicles (NIS, 2018). To this number can also be added the contribution of transit vehicles not



adequately serviced by an external ring road (Fig. 3c) or vehicles of numerous commuting persons from nearby counties or students. In the absence of efficient urban development and mobility measures, in combination with mentality issues (the self-requirement to own and use a car), the city faces regular traffic jams, being ranked as Europe's number 1 capitals (and 5[th] in the world in 2017/11[th] in 2018) when it comes to typical congestion level (TomTom, 2018; typical traffic examples in Fig. 3d,

3e and 3f). Beside traffic, Bucharest's road network health status is precarious, with many dysfunctions related to the quality of embankment, bridges, over or underpasses (Fig. 3a), poor repairing works, limitations in the full utilization of road's length due to illegal (and unsanctioned) parking in many cases (Fig. 3b) or constantly exceeded deadlines for repair or new road works. Another important aspect is that many buildings, not solely in the city center, are highly vulnerable to earthquakes (Toma-Danila et al., 2017): more than 31430 residential buildings were constructed prior to 1946 (294 having more than 4

storeys - a vulnerable category due to long fundamental periods of intermediate-depth Vrancea earthquakes) and 26349 (237 having more than 4 storeys) were constructed between 1946 and 1960 (according to the 2011 National Population and Housing Census), without any compulsory seismic design code, passing through at least one major earthquake with limited evaluation and consolidation afterward (Georgescu and Pomonis, 2018); just 1% of their complete or partial collapse could clearly lead to many deaths and injuries, difficult to manage considering hospital capacity and equipment, as the recent Colectiv Club fire

disaster proved (Marica, 2017), but also to severe road blockages. In 1977, central boulevards (such as Magheru) were closed for at least one week after the 4 March earthquake, still the typical traffic was not severely affected due to low traffic values and the wide use of public transport in those days. Considering the nowadays much wider expected damage scale (Armas et al., 2016; Pavel and Vacareanu, 2016), emergency interventions will have to be provided from multiple locations - inside and outside the city, and usual traffic patterns (not to mention the ones right after a major earthquake, depending also on the time of occurrence) will clearly act against proper reaction. All these problems make Bucharest a highly representative test bed for

the methodology proposed in this article, providing a better understanding of indirect seismic risk due to limitations in road network serviceability capabilities. Pilot analysis for the city where performed in the recent years, using slightly different approaches (Toma-Danila, 2018; Ianos et al., 2017), not so flexible and not at full city scale, but just for the city center. Our results represent the first road network seismic implication analysis for entire Bucharest, offering an important support for

emergency management preparedness (being already considered in the SEISM 2019 exercise) and for risk analysis.©
The starting point for the analysis was the development of a road network GIS database, respecting connectivity and elevation rules. Currently, an official database of such kind is not available for Bucharest. That is why we used OpenStreetMap (OSM) data, which is a very well updated and representative datasource worldwide and for Romania, thanks to the involvement of many local volunteers. OSM road vector data was downloaded using the Geofabrik GIS Data Portal

(http://download.geofabrik.de), requiring additional processing in ArcGIS Desktop's ArcMap (using also Network-risk toolbox), in order to convert it in the ArcGIS network format, accounting for connectivity, hierarchy, travel direction (From-To - FT and To-From - TF), Z-elevation (creating distinctions between roads at ground level, bridges or underpasses) and travel time. For Bucharest - up to the external ring road and its connections to city center, the final number of individual road segments resulted (everything represented in Fig. 3) was 50412. We used data from September 2016; since then up to



December 2019 no major road network modifications occurred (the main exception being the extension of A3 up to north-eastern Bucharest, but with no major influence on our analysis). When analyzing statistics (especially road length) it is important to account for road segments difference of drawing roads per lane or as a whole - otherwise the real number of kilometers will in some cases be doubled. That is why we prefer not to present statistical road length graphs. In the process of defining the rules for the Network Dataset (ND), the Network-risk toolbox requires to add more evaluators beside the ArcGIS

Network analysis extension defaults, the most important being for the obstructions (if a road segment is affected, according to the scenario, the service area analysis using as impedance this evaluator will reveal inaccessible road areas), and other being for different typical traffic scenarios or for economic costs.

For determining (also in a Monte Carlo manner) which road segments can be affected by earthquakes - directly or indirectly, losing connectivity properties, we used the procedures described in Table 1. These account also for the probability of damage

- important when performing Monte Carlo simulations. For this article we performed only 20 Monte Carlo simulations (each taking on an average desktop computer 12 minutes - from simulation to service area results), considered however enough to reflect the damage patterns for the rather extended road network of Bucharest. After 20 runs we observed a stabilization of the results (showed also when comparing aggregated results from 10, 15 or 20 scenarios).

For this study, our approach to account for traffic - representing as we showed a major vulnerability for Bucharest, was based

on replicating typical patterns reflected by Google Traffic, for various representative scenarios:

-   Monday 2AM - no traffic

-   Monday 8AM - morning traffic

-   Monday 6PM - end of work traffic.

Traffic values were obtained by digitizing areas described qualitatively in Google Traffic (very slow, slow, moderate or fast

traffic), identification of roads in these areas (also considering FT and TF ways), modification of travel times (for fast traffic - using the maximum allowed speed, for very slow traffic - 2 km/h), validation with the Google Traffic Direction service (for representative routes crossing the city) and corrections applied in areas with a considerable deviation from the expected values. Although time consuming, this procedure yielded good results. Giving that our analysis focuses on the intervention of emergency vehicles, the influence of traffic lights was neglected (although it can be considered in other analysis purposes) and

the travel speed was considered 50 km/h for fast traffic road segments. For regional and national studies, detailed traffic values might not be needed, since many highway or inter-city roads (generally not crossing urban areas) don't have typical traffic jams impacting furthermore emergency management intervention times.

For estimating post-event traffic patterns, we used a simplified approach, using the following traffic modification parameters:

-   For areas closer to 100 meters (calculated on roads as service area, not as buffer): 2 km/h

-   For areas closer to 500 meters: 5 km/h.

This approach has obvious limitations and uncertainties; however, it provides a flexible and easy-to-compute method of accounting for traffic shifts right after an earthquake, following the findings of Zanini et al. (2017). Modeling traffic driver individual behavior, also over time, is a next step which we will integrate in future studies, also trying to create the means for





validation (recording the traffic patterns after major earthquakes affecting Bucharest or after local incidents in the area of
vulnerable buildings).

In order to enable service area analysis for emergency intervention, hospitals and fire stations were used as facilities. We identified all representative locations in Bucharest and nearby (not including children emergency hospitals, therefore the analysis can be considered as relevant for the adult population). Although of high importance, we could not include for the moment data regarding the capacities of each facility (for example: number of ambulances, hospitals treatment capacity or fire
engine's equipment); these can be considered, reflecting limitations or restrictions in the emergency intervention process (example: how many addresses can be reached within an amount of time due to vehicle availability, how many people can be transported to and hosted by a hospital or where are vehicles with ladders, necessary for intervention in areas with high rise buildings). We will address these issues in further studies when more complex data becomes available; ArcGIS Network Analyst extension can easily accommodate such information and also special evaluators can be added.

The main toward-risk analysis for Bucharest are represented by service areas for emergency management facilities (ambulances for emergency hospitals and fire engines), reflecting which are the times of intervention right after a major earthquake affecting Bucharest (the limit state design earthquake), at three different times for which traffic values are considered. Results can provide a check upon the capabilities to offer a golden hour in medicine (Lerner and Moscatti, 2001) fit intervention - when emergency treatment is most likely to be successful. Also, we analyzed the pre and post-earthquake
time differences for representative economic transit routes, through closest-facility analysis. For these, we used the analysis parameters provided in Table 2. For a single scenario, including Service Area creation, the running time on an average performance desktop PC is less than 10 minutes, given the dense road network of Bucharest.

## 3.2 Results and discussion

In the following figures we present just some of the results obtained for the multiple Monte Carlo and worst-case scenarios
run with Network-risk toolbox (described in Table 2), but also an example aggregation methodology which we used for creating a final index of vulnerable road accessibility for Bucharest.

Fig. 5a and 5b reflect differences between considering all potentially blocked roads and bridges affected and results from Monte Carlo simulations; therefore, Fig. 5a presents for some areas slightly more increased intervention time values. Figure 5c shows service area intervals when considering only emergency hospitals in category I of importance; it can be seen that
their distribution is generally satisfactory, however there is a gap, reflected also by Fig. 5a and 5b, in the south-west area of Bucharest (Rahova and Ferentari neighborhoods) - an area known also for its socio-economic vulnerability (Armas et al., 2016). Due to the significant damage expected in the central area, intervention times are expected to be considerable (given also the traffic values for the considered scenario). The impact of a central hospital such as Coltea, who has currently limited resources to treat the large expected number of patients affected by a major earthquake in the city center is reflected in the
partial decrease of ambulance intervention times for city center, but in the post-earthquake chaos, especially if the earthquake will strike at rush hour, traffic jams are going to pose a considerable threat; our framework partially shows these effects. Bridge



dysfunctionalities do not pose great influences (when comparing also with no damaged bridge scenarios), since in general there are many nearby alternatives; Basarab Overpass (north-west to the center - labeled in Fig. 3) is the only one who could lead to considerable increase of intervention times. Figure 5d is, although difficult to comprehend at first sight, important since

it provides a visual check upon the correlations between minimum intervention times and the number of hospitals who provide this time; if an area is colored towards green and is also hatched, this means that the area is close to multiple emergency hospitals, ensuring a low vulnerability in case of medical emergencies. Data behind this type of maps add an additional understanding to the overall accessibility analysis.

Figure 6 shows service area results for fire stations; the distribution of fire stations is more symmetrical in Bucharest then the

distribution of hospitals, also with a unit in the city center ("Mihai Voda" fire department), behind the Bucharest City Hall building. For this particular scenario (Monday 8AM typical traffic), a good influence of this distribution can be seen south of Union Square (Fig. 6b zoom map), were also boulevards are not expected to be blocked by debris, but north - toward University and Romana Squares, post-earthquake congestion and road segment blockages are expected to significantly increase the time needed to reach the hotspots. To help in the effort of reducing the intervention times in the central area, the "Victoria Palace"

(Government's building) fire department could contribute (if the procedure and situation allows), however we did not find appropriate at the moment to consider it in the analysis.

The total amount of service area maps resulted (for all Monte Carlo scenarios) is considerably large and not relevant independently (the map by map evaluation is more important for quality check and in order to see the stabilization of result patterns). That is why it is needed a further procedure for aggregating data - as it happens with big-data. Providing a data

synthesis easier to grasp, but also showing uncertainties, is also very important for stakeholders. In this purpose we developed a procedure (and a model in the Network-risk toolbox), based on the following reclassification and aggregation procedure:

a) Reclassification of service area polygons for post-earthquake scenarios, according to Table 3.

b) For each service area polygon with identified number of facilities providing the best and second-best intervention time: determination based on Eq. (2) of a counter (C1) reflecting the dependency to a specific facility.

$$C1 = Ni + 0.5 * Ns, \tag{2}$$

where Ni = number of facilities providing the best intervention time; if the service area polygon ≥ 30 minutes, Ni = 0; Ns = number of facilities providing the second best intervention time.

c) Determination of an index (Vi) for each scenario, reflecting the reclassified vulnerability, applied to all polygons, following the considerations in Table 4.

d) Weight overlay of Vi values calculated for emergency hospitals and fire stations, for a specific scenario, applying 25% (0.25) for emergency hospitals and emergency hospitals in category I of importance (in order to reflect the contribution of truly important hospitals in emergency situations), and 50% (0.5) for fire stations (in Bucharest it is relevant to have an important weight for fire stations since they do not provide just equipment for fire extinguishing, but also Mobile Services for Emergency, Reanimation and Extrication, abbreviated SMURD units), leading to a new final vulnerability index per

scenario: Vf.



e) Averaging of Monte Carlo scenario simulations with Vf values.

f) Further averaging of resulted maps with Vf values (6 in total for Bucharest: 3 for the worst-case model and the three traffic scenarios, 3 for Monte Carlo averaged scenario results) for a final results map, revealing the combined index of vulnerable accessibility (Fig. 7).

Figure 7 is the first map of this kind for all of Bucharest, but uncertainties and limitations incorporated in its development are necessary to consider (whether regarding typical traffic scenarios considered, the limited dataset regarding buildings which could collapse during an earthquake or the post-earthquake traffic patterns). The map reflects some of the expected features: a high accessibility vulnerability in central area of the city, due to vulnerable buildings and not so far but difficult to reach in case of an earthquake hospitals (especially in Category I of importance) and fire stations. Also, it shows another area hard to

reach by emergency vehicles: western Bucharest. The map also shows areas with good accessibility (the inner green belt north to the inner ring road) - which are close to hospitals and fire stations and are not considerably influenced by traffic and disrupted road segments.

Another important result is the one presented in Fig. 8. By merging polygons representing areas which can become inaccessible after an earthquake due to road blockages (for each simulation) and also accounting for the number of times these polygons

are generated, a very useful representation of areas difficult to reach can be generated. By using reclassification (in our case based on 5 equal intervals), a qualitative probability for areas to become inaccessible can be expressed. Areas with the lowest probability are generated just for the worst-case model, not appearing during the Monte Carlo limited number of simulations. As expected, inaccessible areas are mostly in the city center (streets such as Blănari, Lipscani, Șelari, Smârdan, Sf. Dumitru, Franceză, Tonița, Eforiei or Biserica Doamnei), where many buildings are expected to block roads and detour routes to the

locations. Other blocked road segments with lower probability could be on streets such as Bărăției, Pătrașcu Vodă, Vasile Lascăr, Poiana Narciselor, Dr. Vasile Sion, Ion Brezoianu, Tudor Arghezi, Batiștei, Jules Michelet etc. Due to the algorithm for Service Area computation, some areas between roads are colored as being blocked (as in Cismigiu Central Park for example), however this is a method limitation and can be eliminated through clipping.

A different product which can be obtained using the network database and the Closest Facility Analysis with emergency

hospitals or fire stations as destinations and high-risk buildings as origins are maps such as the ones presented in Fig. 9. They were obtained by combining the fastest routes for OD pairs, for a given scenario, and show which roads are vital in an emergency situation (need to remain functional since they are critical, providing the quickest access time in the origin). This analysis also shows which hospitals would be preferred (based on adjacency - no medical capabilities are considered) to orient patients to - setting premises for a better preparedness of hospitals expected to have a high patient demand (medical supplies,

hospital beds, doctors etc.).

The seismic risk due to road network dysfunctionalities can be expressed not just by considering the impact of road blockage and traffic on emergency intervention, leading to time limitations in reaching patients. When roads are closed, connectivity throughout the city can be lost for days, weeks or years, with a high impact on economy - due to delays in stock supplies and production, greater costs for carburant or loss of clients. The created dataset can be used also to monitor which are the



differences between pre and post-earthquake travel times, for representative OD pairs. For this study we selected 8 pairs relevant to cardinal points, some with links to the city center and some aimed to show if in case of an earthquake the initially preferred route throughout the city is going to change in favor of the external ring road. Results are presented in Table 5 and in Fig. 10. For the 2 AM traffic scenario, differences are not significant, however for the others - especially for routes which need to reach the city center (Piata Universitatii for example), there are clear values showing a mean travel time increase from
110-120% to 300-432%, for the Centura (external ring road) - Otopeni -> University Square route.

## 4 Conclusions

In this paper we presented a new methodology for evaluating of direct and indirect implications of natural hazards on transportation network, generally applicable and adaptive to various types of hazards, networks or data availability. Starting from structural evaluation, the analysis focuses on systemic or functional assessment, expressing furthermore the risk due to
lack of connectivity, for example. After determining exposure, vulnerability and hazard factors, leading to the definition of the network and the identification of segments which can become unusable (and the probability of this to happen), the methodology can perform Monte Carlo simulations resulting in multiple scenarios evaluated individually in terms of generated risk (for emergency intervention or socio-economic aspects) and aggregated into final risk indexes. There are also capabilities on accounting for pre and post disaster traffic and for emergency facilities capacity or equipment. In order to facilitate the use of
the methodology we also integrated it into an open toolbox (collection of models) - free to download and customize, entitled Network-risk (available on www.infp.ro/network-risk). This toolbox is for now dependent on the geoprocessing algorithms implemented in the widely used commercial software ArcGIS Desktop Advanced, with Network Analyst extension. In the near future we will try to integrate Network-risk also in non-commercial GIS software such as QGIS, who still require at the moment more development toward advanced network analysis. Network-risk toolbox is under continuous development and in
future versions more features will be available, so please check regularly the website.

To prove its capabilities, Network-risk was tested on the entire road network of Bucharest, Romania, one of Europe's most endangered capitals due to earthquakes, considering the high seismic hazard of intermediate-depth Vrancea earthquakes, the vulnerable building stock (349 high or moderate rise buildings are categorized in the seismic risk class I in January 2016, representing just the tip of the iceberg) but also major traffic congestion. One of the most difficult parts in the analysis is the
proper input data collection. As we showed, this can be achieved (at least for a preliminary form) more easily - by using OpenStreetMap data along with a Network-risk module designed to arrange (partially automatically) the network data into ArcGIS format, digitized traffic areas based on Google Traffic layers or empirical formulas, literature fragility functions and expert judgement for determining road segment failure probabilities. Our analysis focused both on the evaluation of emergency intervention times, for emergency hospitals and fire stations and on the evaluation of economic implications for representative
commercial routes (time delays in post-earthquake conditions). Results show that the city center would be significantly vulnerable not just because of collapsing buildings but also due to the difficulty to reach these sites by ambulances and



firefighters - although there are facilities nearby, such as the Coltea Hospital (however not of category of importance I) and the "Mihai Voda" fire department; due to road blockages and traffic jams, considering the Monday 8AM and 6PM typical traffic scenarios, these could have quick access just to a few surrounding streets. The aggregated results show that also for the western parts of Bucharest intervention times could be significant. When calculating service areas, we believe that our approach yielded better results due to considering the dependency to a single facility to provide the minimum intervention time. Results are very useful for determining new locations for emergency facilities, for increasing capacities in facilities near vulnerable areas or for traffic management planning. The service area analysis of emergency hospitals also shows the necessity of such a hospital in the south-western part of Bucharest - an area also known for its high socio-economic vulnerability (Armas et al., 2016). For the city center, a strategy in case of an earthquake has to be elaborated and put into place, referring to facilitating access in the area, traffic redirection and design of safe road access corridors; considering also that the vulnerability of routes connecting the city center especially with north or south destination is significant, with travel time increase greater than 150% in typical scenario conditions. Since there was no capability and relevance, considering previous earthquakes (the ones in 1940 or 1977), we will also be working on creating the means for result validation, as for Bucharest a major earthquake is certain to happen and new devices and technologies are certain to record what happens to traffic and road blockages. We will also be testing Network-risk for regional analysis, using also rapid seismic loss estimations generated by the Seisdaro System of INFP (Toma-Danila et al., 2018).

We hope that this article will provide researchers important guidelines on how to analyze the risks of transportation networks affected by natural hazards and a practical tool to be applied in other parts of the world and stakeholders an example of useful results which they could benefit from, in their efforts to understand and mitigate risks.

**Code and data availability**

The Network-risk toolbox for ArcGIS Desktop and sample data for Bucharest (used for this study) can be downloaded, with user manual, at www.infp.ro/network-risk. Please revisit the address and check for new versions, since the toolbox is constantly being upgraded.

**Author contribution**

DT-D and IA developed the methodology and DT-D and AT implemented and tested it in GIS, obtaining results analyzed also by IA. DT-D prepared the manuscript with contributions from all co-authors.

**Competing interests**

The authors declare that they have no conflict of interest.



**Acknowledgement**

This work was financed through the doctoral scholarship of the main author, from the University of Bucharest, Faculty of Geography, and from the Seismology and Earthquake Engineering Research Infrastructure Alliance for Europe (SERA) project - EU Horizon 2020 Programme, Grant No. 730900.

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

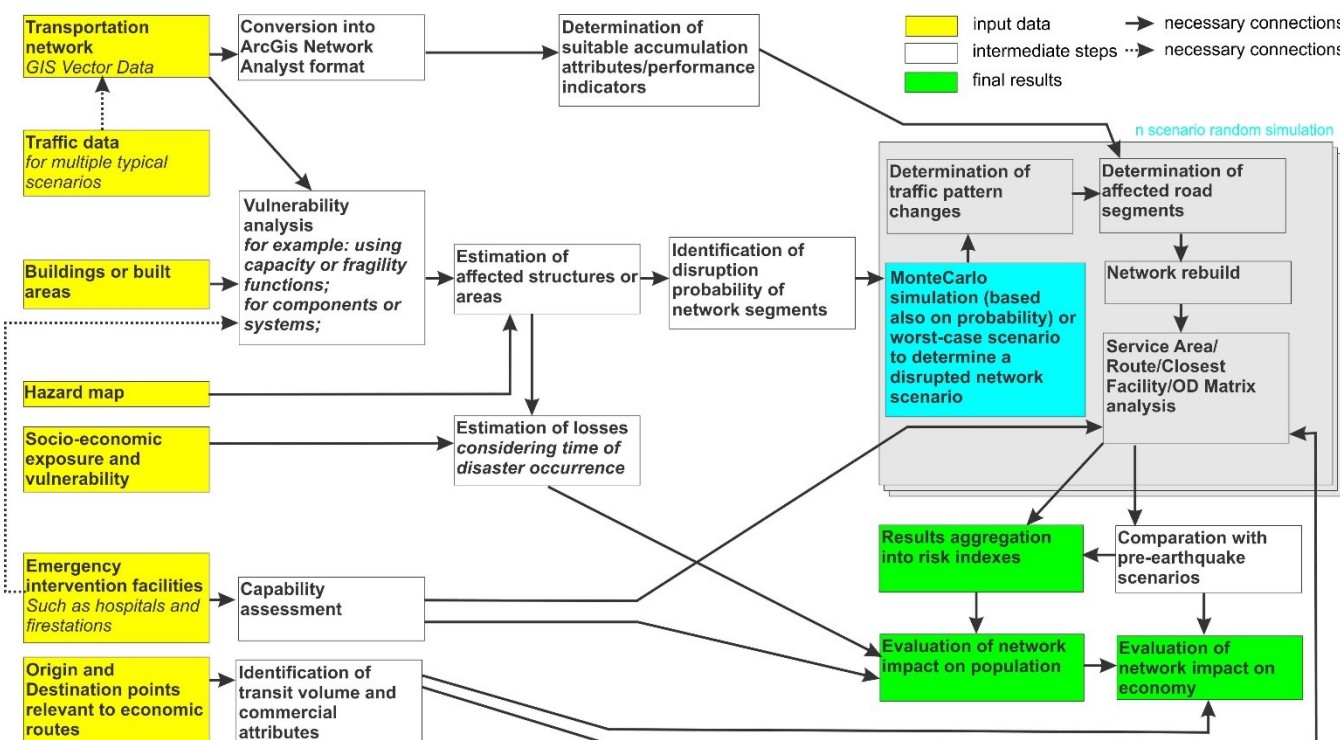

**Figure 1: Graphical representation of the proposed methodology for evaluating the implications of transportation network damage**
**due to natural hazards, integrated in the Network-risk toolbox.**



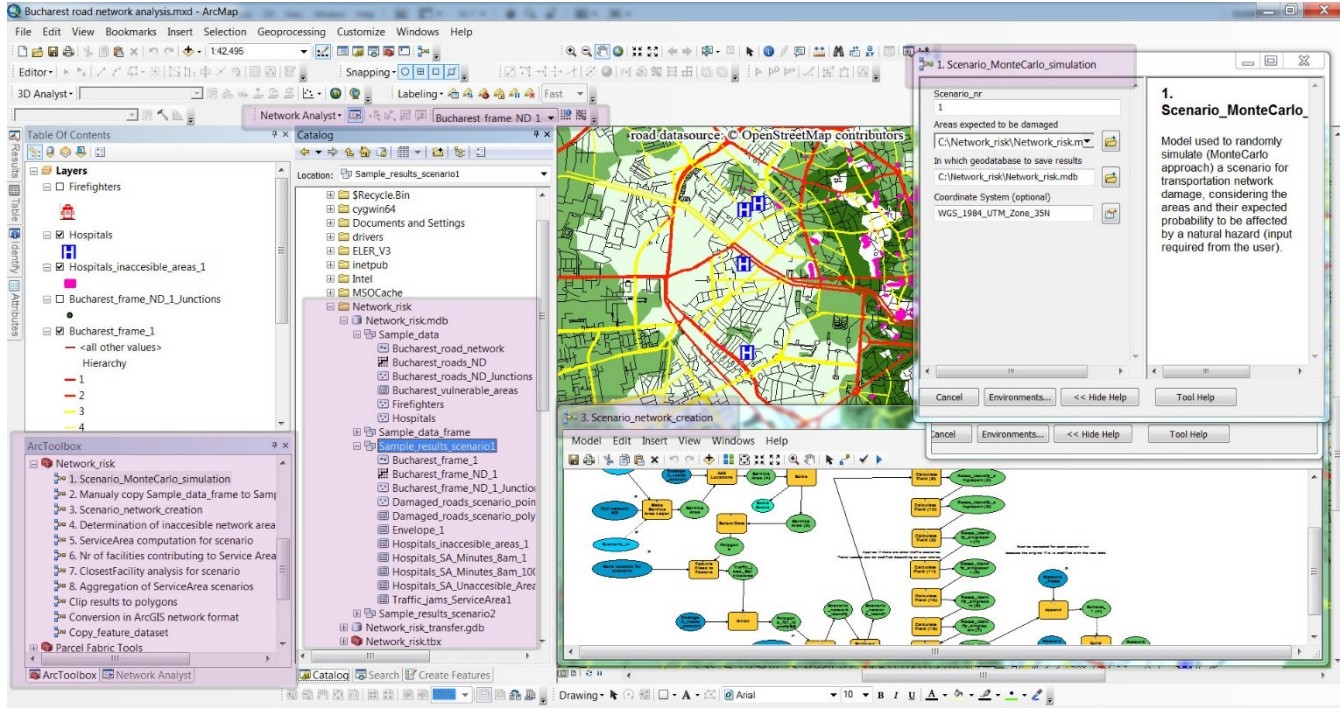

**Figure 2: Screen capture of ArcGIS Desktop ArcMap with Network-risk toolbox added, contributing to the analysis of Bucharest's road network risk analysis; the framework of one of the models (3. Scenario network creation) can be seen, as well as the model run interface (1. Scenario Monte Carlo simulation), the Network-risk toolbox modules and the sample data results created using these modules (highlighted with purple).**





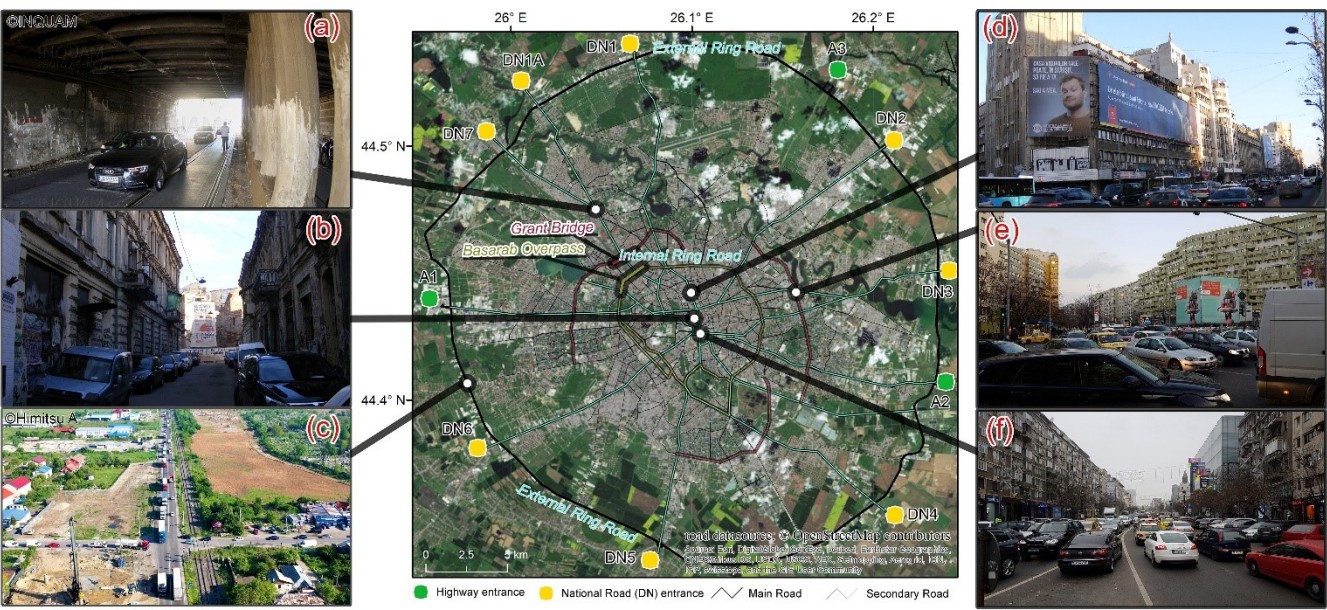

**Figure 3: Bucharest road network map, highlighting main roads and connections with highways and national roads (datasource: © OpenStreetMap contributors 2019. Distributed under a Creative Commons BY-SA License, data from September 2016; basemap: © ESRI and contributors), and examples of vulnerable underpasses [(a) Constanta Bridge, © INQUAM], illegal parking (b), southern External Ring Road disfunctionalities [(c) © Himitsu A.], typical rush-hour traffic (d, e, f) and vulnerability due to old buildings with seismic risk (b).**



**Table 1: Factors considered for determining the probability of road segments to be affected by earthquakes.**

| Factor | Method of analysis |
|---|---|
| Bridges | Mean fragility functions from Crowley et al. (2011), for the corresponding structural typology (mostly reinforced concrete) were used. Considering the microzonation map of Marmureanu et al (2010) for maximum PGA values in Bucharest due to the largest probable earthquake in Vrancea, the complete damage probabilities obtained were small: 1.5 - 2%. For the Basarab Overpass fragility functions were adapted due to different characteristics (suspension and steel arch bridge sections, seismic passive dampers), considering descriptions in Sartori M. (2012). |
| Roads blocked by building debris | We used Eq. (1) (from Moroux et al., 2004) to determine the probability of roads to be blocked by the debris generated by the collapse of buildings in the seismic risk class I, which are most likely to collapse during the limit state design earthquake (349 in total, mostly with more than 4 storeys, according to Bucharest City Hall data from January 2016 - https://amccrs-pmb.ro/liste-imobile); the footprint of buildings was determined, and buffers were added according to debris area; the output (Fig. 4) was supplemented by expert judgement based on satellite images, building structural considerations and building vicinity, road width etc., to attribute road blockage probabilities - ranging from 1 to 70%, since no building is certain to collapse.<br><br>$$\text{Debris area (meters)} = \frac{2}{3} * \text{Number of floors} \qquad (1)$$ |
| Liquefaction | We attempted to use some data (Neagu et al., 2018), but eventually the liquefaction map was considered to generic; after more detailed analysis we can integrate it into the analysis. |

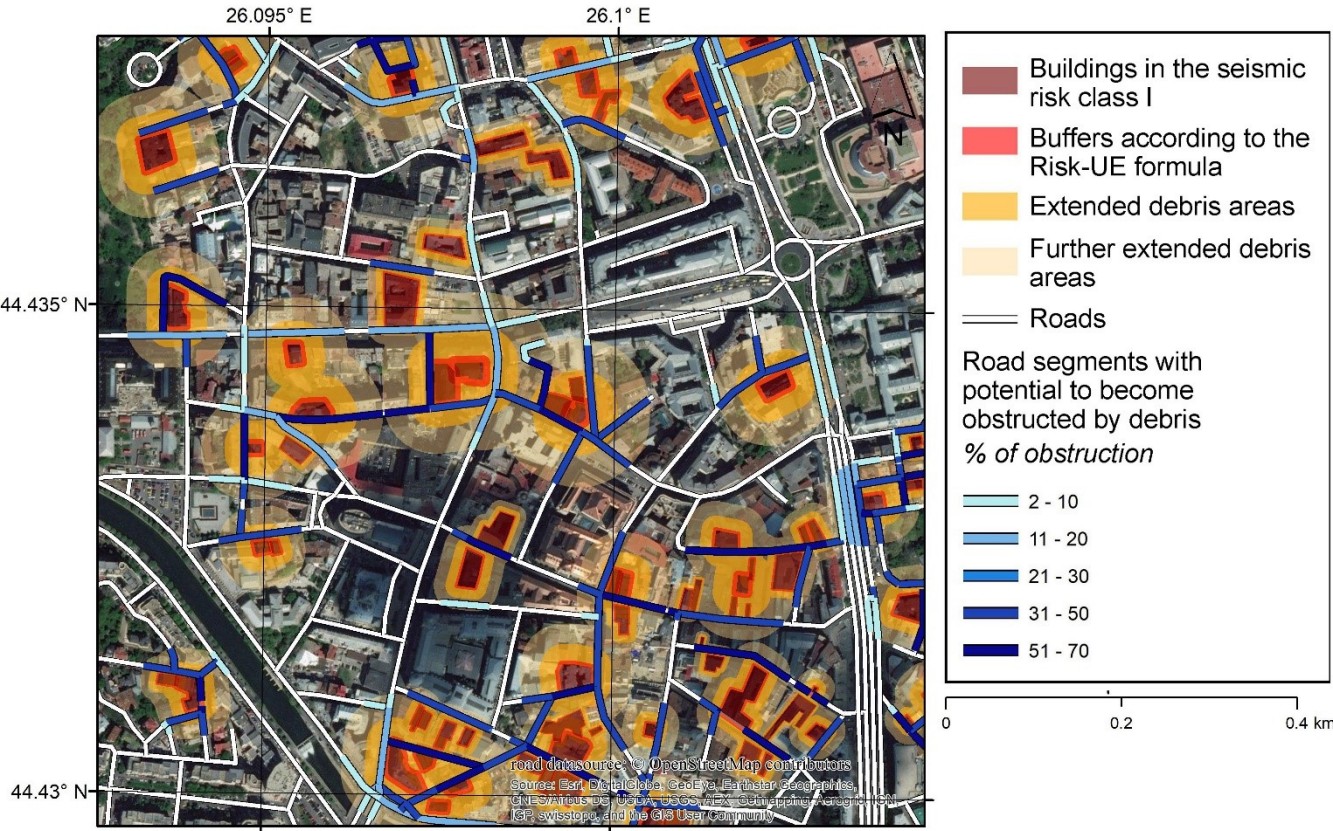

**Figure 4: Example of road blockage analysis due to building debris, applied for the historical center of Bucharest (datasource: © OpenStreetMap contributors 2019. Distributed under a Creative Commons BY-SA License, data from September 2016; basemap: © ESRI and contributors).**









**Table 2: Parameters used for of Bucharest post-earthquake road network risk analysis.**

| Facilities | Analysis parameters | Considered scenarios | Number of maps resulted |
|---|---|---|---|
| Emergency hospitals | Analysis type: Service Area<br>- impedance attributes: minutes (depending on traffic scenario);<br>- Default breaks: 5, 10, 15, 20, 25, 30, 35, 40, 45, 50, 55, 60 minutes;<br>- no one way restrictions (emergency management vehicles are allowed not to respect these restrictions); | For three traffic scenarios (2AM, 8AM and 6PM) - pre and post-earthquake, considering 20 Monte Carlo scenarios and the worst-case model (failure of all listed segments) for blocked roads and bridges (which for Bucharest have a very low damage probability, and that is why for our worst-case simulations we made a custom selection based on their health condition, year of construction and length). | 3 (pre-earthquake) + 9 (post-earthquake - worst-case model, including analysis of facilities which provide the number best and second-best times for intervention) + 20 (post-earthquake, Monte Carlo scenarios) + 3 (post-earthquake, Monte Carlo averaged scenarios) |
| Emergency hospitals in category I of importance (since they have the main capacity and responsibilities in case of an earthquake) | - travel from facility;<br>- restrictions: polygon barriers (inaccessible areas provided by identifying holes from initial Service Area analysis using the "obstruction" column as impedance attribute | | 3 + 9 + 20 + 3 |
| Fire stations | - module provided in the Network-risk toolbox). | | 3 + 9 + 20 + 3 |



| Facilities | Analysis parameters | Considered scenarios | Number of maps resulted |
|---|---|---|---|
| Origin-destination pairs for representative economic transit routes | Analysis type: Closest Facility<br>- impedance: minutes (depending on traffic scenario);<br>- Facilities to Find: the total number of origins/destinations (to be able to extract not just the statistics as with Cost Matrix analysis, but also the path of the route).<br>- accumulators: minutes (depending on traffic scenario) and meters;<br>- analysis was performed also by changing initial origins within destinations (to show differences due to traffic ways and one-way restrictions). | For three traffic scenarios (2AM, 8AM and 6PM) - pre and post-earthquake | 3 (pre-earthquake) + 3 (post-earthquake - worst-case scenario)<br>+ 3 time difference tables |





**Figure 5: Service Areas for emergency hospitals, for the Monday 8AM typical traffic scenario and for: (a) the worst-case model; (b) a Monte Carlo scenario; (c) emergency hospitals in category I of importance and the worst-case model; (d) the number of emergency hospitals providing the minimum intervention time in the worst-case model (datasource: © OpenStreetMap contributors 2019. Distributed under a Creative Commons BY-SA License, data from September 2016).**
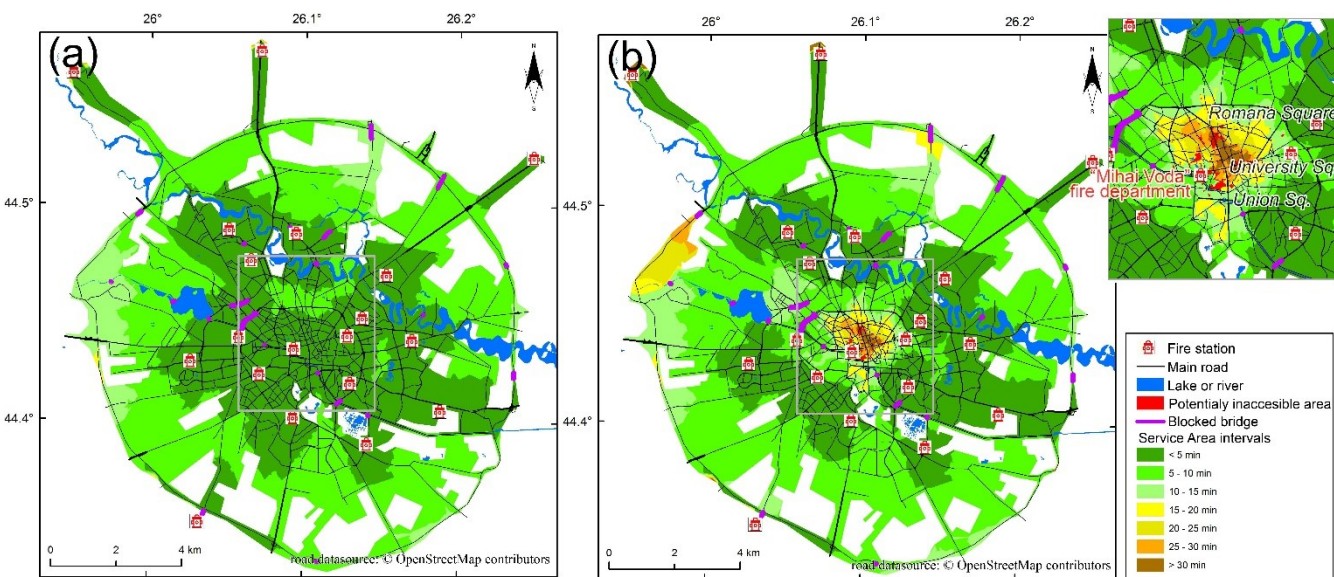

**Figure 6: Service Areas for fire stations, (a) pre-earthquake and (b) post-earthquake, considering the worst-case model, for the Monday 8AM typical traffic scenario (datasource: © OpenStreetMap contributors 2019. Distributed under a Creative Commons BY-SA License, data from September 2016).**

**Table 3: Reclassification intervals for service area polygons.**

| Default breaks for service areas | Reclassification values (Vr) |
|---|---|
| ≤ 10 minutes | 1 |
| 10 - 15 minutes | 2 |
| 15 - 20 minutes | 3 |
| 20 - 25 minutes | 4 |
| ≥ 25 minutes | 5 |




**Table 4: Formulas for calculating the index for reclassified vulnerability (Vi); C1 intervals are relative to the facility database and study area characteristics.**

| Formula for Vi | Conditions - depending on C1 values |
|---|---|
| Vi = Vr - 0.5 | if C1 >= 5 for emergency hospitals and fire stations |
|  | if C1 >= 3 for emergency hospitals in category I of importance |
| Vi = Vr (applied also to scenarios without calculated C1 values) | if 2 >= C1 < 5 for emergency hospitals and fire stations |
|  | if 2 >= C1 < 3 for emergency hospitals in category I of importance |
| Vi = Vr + 0.5 | if C1 < 2 for emergency hospitals and fire stations |
|  | if C1 < 2 for emergency hospitals in category I of importance |



**Figure 7: Final map showing qualitative values for the combined final index of vulnerable road network accessibility (Vf) for**
**Bucharest (datasource: © OpenStreetMap contributors 2019. Distributed under a Creative Commons BY-SA License, data from September 2016).**

**Figure 8: Areas who can become inaccessible immediately after an earthquake (datasource: © OpenStreetMap contributors 2019. Distributed under a Creative Commons BY-SA License, data from September 2016; basemap: © ESRI and contributors).**

Natural Hazards
and Earth System
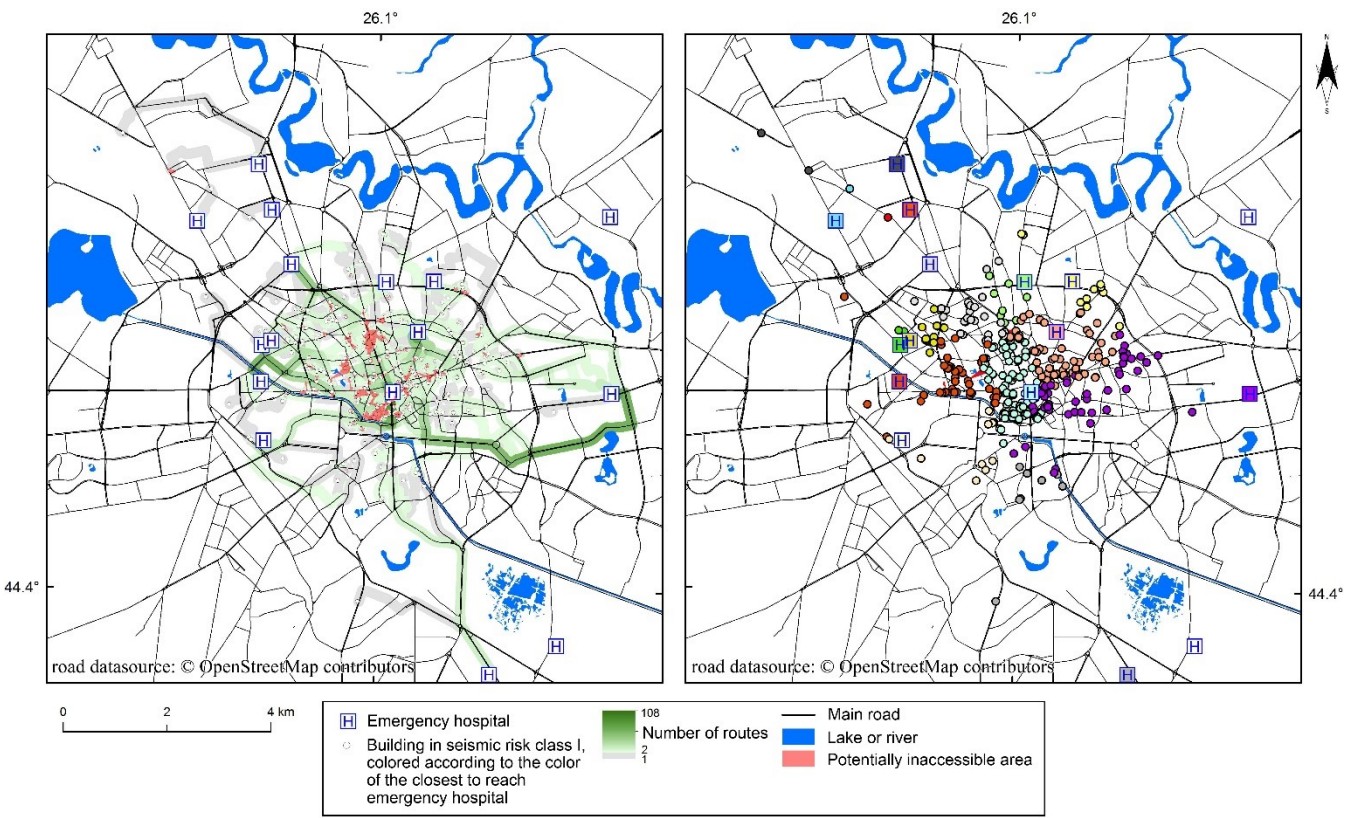

**Figure 9: Maps reflecting fastest routes (and the density of these routes) between buildings in seismic risk class I and (a) emergency hospitals, and (b) the closest hospital, for the Monday 8AM typical traffic scenario (datasource: © OpenStreetMap contributors 2019. Distributed under a Creative Commons BY-SA License, data from September 2016).**

**Table 5: Time differences (expressed in minutes) between various OD pairs shown in Figure 10 and for pre and post-earthquake conditions.**

| Route | From-To (FT), minutes | | | To-From (TF), minutes | | |
|---|---|---|---|---|---|---|
| | 2 AM | 8 AM | 6 PM | 2 AM | 8 AM | 6 PM |
| Centura-A1 -> Piata Unirii | 0 | 25 | 19 | 1 | 24 | 21 |
| Piata Unirii -> Metrou Pantelimon | 0 | 11 | 14 | 0 | 8 | 10 |
| Centura-Otopeni -> Piata Universitatii | 5 | 72 | 77 | 8 | 62 | 63 |
| Piata Universitatii -> Centura-Giurgiului | 1 | 30 | 33 | 3 | 45 | 44 |
| Centura-Chitila -> Centura-Oltenitei | 1 | 0 | 0 | 1 | 10 | 0 |
| Drumul Taberei -> Centura-Splai | 0 | 4 | 5 | 0 | 5 | 4 |
| Centura-Magurele -> Gradina Zoologica | 0 | 18 | 7 | 0 | 1 | 9 |
| Metrou Eroii Revolutiei -> Spitalul Fundeni | 1 | 6 | 11 | 1 | 6 | 10 |


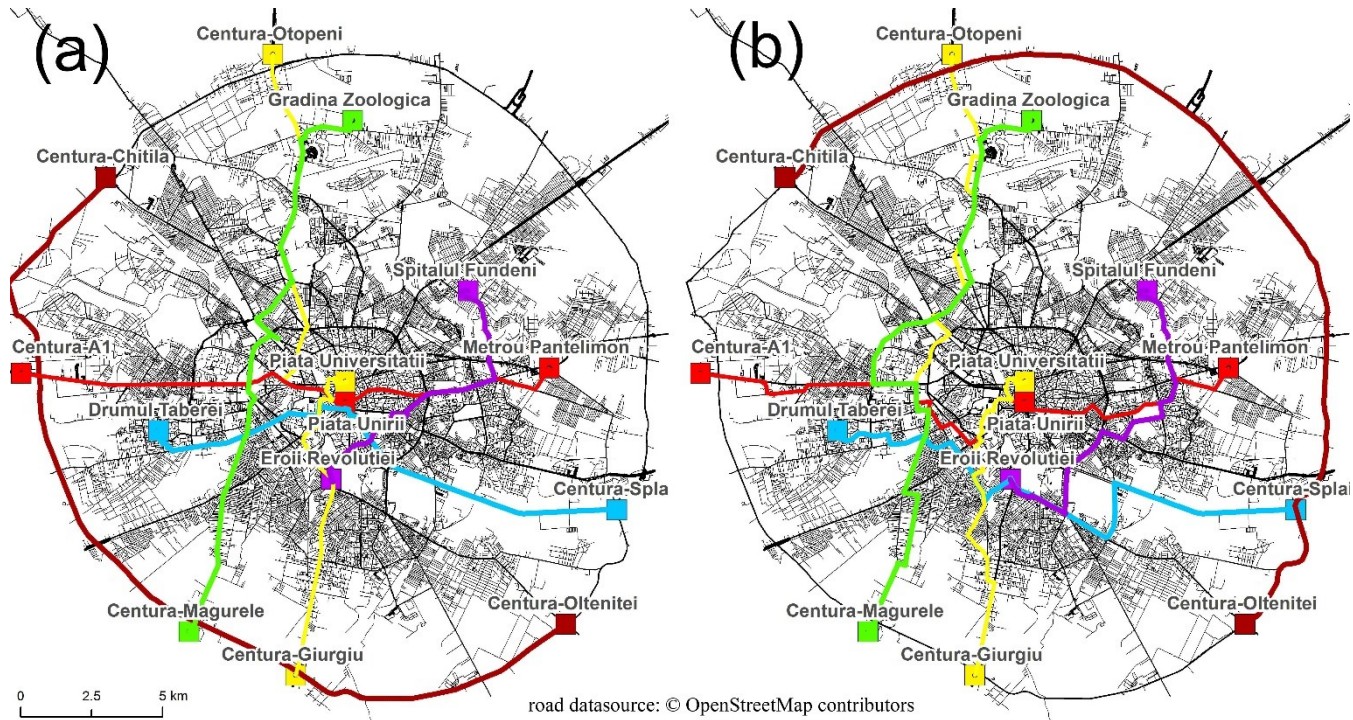

**Figure 10: Fastest routes for 8 representative OD pairs for Bucharest, for the (a) FT directions and for the Monday 2AM and (b) 6PM typical traffic scenario (road datasource: © OpenStreetMap contributors 2019. Distributed under a Creative Commons BY-SA License, data from September 2016).**