# Peer review of "Network-risk: an open GIS toolbox for estimating the implications of transportation network damage due to natural hazards, tested for Bucharest, Romania"

_Natural Hazards and Earth System Sciences, 2019_

## Referee Comment (RC1) · Anonymous Referee #1 · 20 Jan 2020

Overall, this is an important paper on earthquake exposure risk of road networks in Bukarest. The novelty being an up to date study on this earthquake prone city - important also for international fellow researchers to compare their approaches and advance. Very explicit and useful maps, and tables making their approach transparent.

Language should be edited by a native speaker or professional editor.

Minor comments: Page 1 Line 26: More recent source than 2008 would be good to add

Section 2 Line 99: I suggest avoiding suggestive expressions such as "is at first sight easy to follow" I guess, an -s is missing for "comprise"

Line 107: This needs more detailed explanation - what exactly is novel here? Level of serviceability, random network analyses etc already do exist?! The following sentences up to line 116 are fine, but still, what is novel about this? Maybe: few conducted studies on multiple hazards and specifically, in Bukarest...

Paragraph around line 125: reconsider wording, length of sentences and maybe, grammar. Also further down below, for instance, Lines 153-154 You state "A probability of 100% for a network segment would indicate certain blockage - very hard to consider for a transportation network, but worst-case scenario could use this value." What about single roads, dead end roads, last road segments before a harbour, hospital emergency entrance, airport etc.? Many more section, just an example: Line 190

What is missing a bit in section 2 is discussion of alternative models, such as QGIS, GRASS or A* algorithm - but mights also be taken up in the discussion chapter. So far, it looks rather that Dijkstra was selected as only available algorithm.

Regarding the structured research questions (which I think are fine by themselves) it could be reconsidered how much they fit to a) section 2 and b) the following assessment. For section 2, it would be good to know what the authors consider as "vulnerability" (of the road, of users. etc.), and "socio-economic" - why were no research mentioned on this specifically before to guide the reader that this aspect is in fact most relevant. And to the very last question segment - maybe add a methodology short description of state of the art location-allocation analyses?

It is a bit unusual to have methods both in 2 and 3.1 sections - maybe reconsider merging it and separating the results? 3.2 I suggest starting not with such a detail sentence but rather coming back to the research questions and following their structure, or, the nice structure/steps laid out in section 1.

Section 3.2 is written largely for experts familiar with such methodology, which is fine. However, some more structure could help, such as following the research questions. Also, an intro part easier to understand for non-experts. For the experts then, more explanation on why certain values were decided on such as 30min service area (would that be enough to save lives? (Make sure to match it also with Table 3 - 25mins...)Which sources support this "long" time - being more realistic to earthquake debris routing maybe, but following which estimations, sources, previous studies?). The same for weighting: it is common for modellers to pick weights themselves and the explanatory sentence is fine, however, the mobile rescue teams might not have the same resources treating >1000 wounded and might be blocked by the debris - any sources supporting this?

A discussion chapter is missing - the authors have decided to mix results and discussion / rather commenting of methodological steps. The paper would benefit from a sub-section on shortcomings and recommendations for fellow researchers regarding methodology, maybe also a sub-section how the study matches with current similar studies - or not and provides novelty. Conclusion (and maybe discussion) could also make good use of the structure of research questions.

While the text is well-written and the professionalism of the assessment and knowledge about the literature out of question, the text could use a bit more structure here and there, as in the long section texts in 2 and 3, for example. Setting key terms per paragraph in italics could be an option to guide the reader, or sub-sections, or summarizing flow-charts.

What is missing a bit, at least conceptually, (it must not all be analysed within one paper): the perspective of affected people or customers of roads and logistics/critical infrastructure. They also interact with roads and their usage (see Rinaldi et al. 2001), be it through geographical, physical or logical interaction. The article starts with a good understanding of the recent trend of balancing hazard and vulnerability, but then focuses too much on the exposure only. Maybe it would be sufficient expressing this

demand more in chapter 1 and 2 and ... a bit, then it should be fine.

---

## Referee Comment (RC2) · Mihai Micu (Referee) · 17 Feb 2020

1.   General comments:  the authors are presenting an approach which might have a consistent application (not only for Romania, but worldwide) in terms of exposure/vulnerability/risk analysis.  Moreover, there are numerous stakeholders which may show practical interest in this application, both coming from the prevention/preparedness or response/recovery parts of the risk management spectrum. The manuscript follows a rather clear and logic structure. There are consistent chapters devoted to methodology, results but not so much discussions, overall witnessing a good

knowledge of the authors in both theoretical and applied issues. The manuscript is written in good English (sometimes with long sentences) and the graphic part is (mostly) clear and strongly backs-up the written text. 2. Specific comments: to our opinion, the structure of the manuscript could be improved by rearranging the text according to the chapters. Consistent paragraphs in the Results chapter (e.g. those following line 250) are more fit to the Methodology chapter; meanwhile, at Results there are considerations which we find more suitable for the description of the study area (see Fig.6). There are some references which deserves an update (some 10-15 years old; see lines 26, 58, 82, 152), since in the recent years, similar applications have been developed (see rupok.cz). In our opinion, a consistent part of the discussions should be devoted to the following issue: how useful is such an application and which is its effectiveness? As mentioned by the authors, it is important not for the scientists, but a more consistent part should be devoted to: which is the main outcome - improved exposure analysis or improved vulnerability assessment; how it might improve the cost-benefit analysis if it addresses risk evaluation (as written in the abstract); which is its main applicability - prevention or response (since based on this, different stakeholders should be interested); was any feed-back requested in this respect? In the mean time, the authors are mentioning numerous uncertainties behind such an approach, which brought in the same context with its high applicability, deserve a larger explanation which could rank its effectiveness. 3. Technical corrections: - the graphic part may be improved by replacing some of the written names: Fig.6 - better if the names are in the legend, since on the map they look rather general. - lines 110-111: difficult to understand, is there something missing? "... to be considered" maybe? - line 177: already mentioned; - line 223: an explanation of the statement "very well updated and representative" is needed; - there are names which sometimes are in English, sometimes in Romanian (e.g. Piata Universitatii vs. University Square); they should all follow the same writing.

---

## Referee Comment (RC3) · Anonymous Referee #3 · 26 Feb 2020

a. Considering that Bucharest represents one of the most exposed capitals in Europe to strong earthquakes that has a poor and crowded road infrastructure, this paper represents an important step for future worldwide studies. We consider that the paper is written in good English and that the figures are clear and easy to understand. b. Form our point of view the manuscript could be reorganised a little bit. Some paragraphs fit much better in different sections (e.g. from results moved to methodology). c. We consider that a discussions chapter should be defined in order to explain/ describe the efficiency of the methods and algorithms described in the paper and their associated

errors. d. Some newer references could be added in the review of current studies.

---

## Author Response (AR1)

Overall, this is an important paper on earthquake exposure risk of road networks in Bukarest. The novelty being an up to date study on this earthquake prone city - important also for international fellow researchers to compare their approaches and advance. Very explicit and useful maps, and tables making their approach transparent.

Language should be edited by a native speaker or professional editor.

Minor comments: Page 1 Line 26: More recent source than 2008 would be good to add

[Figure]

Section 2 Line 99: I suggest avoiding suggestive expressions such as "is at first sight easy to follow" I guess, an -s is missing for "comprise"

Line 107: This needs more detailed explanation - what exactly is novel here? Level of serviceability, random network analyses etc already do exist?! The following sentences up to line 116 are fine, but still, what is novel about this? Maybe: few conducted studies on multiple hazards and specifically, in Bukarest...

Paragraph around line 125: reconsider wording, length of sentences and maybe, grammar. Also further down below, for instance, Lines 153-154 You state "A probability of 100% for a network segment would indicate certain blockage - very hard to consider for a transportation network, but worst-case scenario could use this value." What about single roads, dead end roads, last road segments before a harbour, hospital emergency entrance, airport etc.? Many more section, just an example: Line 190

What is missing a bit in section 2 is discussion of alternative models, such as QGIS, GRASS or A* algorithm - but mights also be taken up in the discussion chapter. So far, it looks rather that Dijkstra was selected as only available algorithm.

Regarding the structured research questions (which I think are fine by themselves) it could be reconsidered how much they fit to a) section 2 and b) the following assessment. For section 2, it would be good to know what the authors consider as "vulnerability" (of the road, of users. etc.), and "socio-economic" - why were no research mentioned on this specifically before to guide the reader that this aspect is in fact most relevant. And to the very last question segment - maybe add a methodology short description of state of the art location-allocation analyses?

It is a bit unusual to have methods both in 2 and 3.1 sections - maybe reconsider merging it and separating the results? 3.2 I suggest starting not with such a detail sentence but rather coming back to the research questions and following their structure, or, the nice structure/steps laid out in section 1.

[Figure]

Section 3.2 is written largely for experts familiar with such methodology, which is fine. However, some more structure could help, such as following the research questions. Also, an intro part easier to understand for non-experts. For the experts then, more explanation on why certain values were decided on such as 30min service area (would that be enough to save lives? (Make sure to match it also with Table 3 - 25mins...)Which sources support this "long" time - being more realistic to earthquake debris routing maybe, but following which estimations, sources, previous studies?). The same for weighting: it is common for modellers to pick weights themselves and the explanatory sentence is fine, however, the mobile rescue teams might not have the same resources treating >1000 wounded and might be blocked by the debris - any sources supporting this?

A discussion chapter is missing - the authors have decided to mix results and discussion / rather commenting of methodological steps. The paper would benefit from a sub-section on shortcomings and recommendations for fellow researchers regarding methodology, maybe also a sub-section how the study matches with current similar studies - or not and provides novelty. Conclusion (and maybe discussion) could also make good use of the structure of research questions.

While the text is well-written and the professionalism of the assessment and knowledge about the literature out of question, the text could use a bit more structure here and there, as in the long section texts in 2 and 3, for example. Setting key terms per paragraph in italics could be an option to guide the reader, or sub-sections, or summarizing flow-charts.

What is missing a bit, at least conceptually, (it must not all be analysed within one paper): the perspective of affected people or customers of roads and logistics/critical infrastructure. They also interact with roads and their usage (see Rinaldi et al. 2001), be it through geographical, physical or logical interaction. The article starts with a good understanding of the recent trend of balancing hazard and vulnerability, but then focuses too much on the exposure only. Maybe it would be sufficient expressing this

demand more in chapter 1 and 2 and ... a bit, then it should be fine.

[Figure]

Nat. Hazards Earth Syst. Sci. Discuss.,
https://doi.org/10.5194/nhess-2019-409-RC2, 2020

1. General comments: the authors are presenting an approach which might have a consistent application (not only for Romania, but worldwide) in terms of exposure/vulnerability/risk analysis. Moreover, there are numerous stakeholders which may show practical interest in this application, both coming from the prevention/preparedness or response/recovery parts of the risk management spectrum. The manuscript follows a rather clear and logic structure. There are consistent chapters devoted to methodology, results but not so much discussions, overall witnessing a good

knowledge of the authors in both theoretical and applied issues. The manuscript is written in good English (sometimes with long sentences) and the graphic part is (mostly) clear and strongly backs-up the written text. 2. Specific comments: to our opinion, the structure of the manuscript could be improved by rearranging the text according to the chapters. Consistent paragraphs in the Results chapter (e.g. those following line 250) are more fit to the Methodology chapter; meanwhile, at Results there are considerations which we find more suitable for the description of the study area (see Fig.6). There are some references which deserves an update (some 10-15 years old; see lines 26, 58, 82, 152), since in the recent years, similar applications have been developed (see rupok.cz). In our opinion, a consistent part of the discussions should be devoted to the following issue: how useful is such an application and which is its effectiveness? As mentioned by the authors, it is important not for the scientists, but a more consistent part should be devoted to: which is the main outcome - improved exposure analysis or improved vulnerability assessment; how it might improve the cost-benefit analysis if it addresses risk evaluation (as written in the abstract); which is its main applicability - prevention or response (since based on this, different stakeholders should be interested); was any feed-back requested in this respect? In the mean time, the authors are mentioning numerous uncertainties behind such an approach, which brought in the same context with its high applicability, deserve a larger explanation which could rank its effectiveness. 3. Technical corrections: - the graphic part may be improved by replacing some of the written names: Fig.6 - better if the names are in the legend, since on the map they look rather general. - lines 110-111: difficult to understand, is there something missing? "... to be considered" maybe? - line 177: already mentioned; - line 223: an explanation of the statement "very well updated and representative" is needed; - there are names which sometimes are in English, sometimes in Romanian (e.g. Piata Universitatii vs. University Square); they should all follow the same writing.

[Figure]

Nat. Hazards Earth Syst. Sci. Discuss.,
https://doi.org/10.5194/nhess-2019-409-RC3, 2020

a. Considering that Bucharest represents one of the most exposed capitals in Europe to strong earthquakes that has a poor and crowded road infrastructure, this paper represents an important step for future worldwide studies. We consider that the paper is written in good English and that the figures are clear and easy to understand. b. Form our point of view the manuscript could be reorganised a little bit. Some paragraphs fit much better in different sections (e.g. from results moved to methodology). c. We consider that a discussions chapter should be defined in order to explain/ describe the efficiency of the methods and algorithms described in the paper and their associated

errors. d. Some newer references could be added in the review of current studies.

[Figure]

**Relevant changes made in the manuscript**

- New introductive phrases were added to chapter 3.3 (3.2 in the first version).
- We corrected the values in Table 3 with > 30 min. as maximum reclassification interval
5 - A discussion chapter was added at the end of the article; here there were also added commentaries about methodological shortcomings, recommendations and future plans.
- We have moved and rephrased many paragraphs in the new version of the manuscript, also creating new subchapters, such as 3.1 Case study area description, 3.2 Data and methods considered for Bucharest or 4.1 Discussion.
- We have added two newer and comprehensive references of top institutions:
10      o  Pesaresi M., Ehrlich D., Kemper T., Siragusa A., Florczyk A., Freire S., and Corban C.: Atlas of the Human Planet 2017: Global Exposure to Natural Hazards, Publications Office of the European Union, doi: 10.2760/19837, 2017.
        o  Gu D.: Exposure and vulnerability to natural disasters for world's cities; United Nations Department of Economic and Social Affairs, Population Division, Technical Paper No. 2019/4, 2019.
15      o  Pinto et al. (2012), Sevtsuk and Mekonnen (2012), Gu, 2019; Pesaresi et al., 2017; Vodak et al., 2015; Koks et al., 2019, Jenelius and Mattsson, 2015; Santarelli et al., 2018 etc.
- We have re-read the manuscript and made significant adjustements in order to split long sentences in smaller ones.
- We have added clearer paragraphs such as "Stakeholders such as emergency situations managers provided us important feedback, acknowledging that final products can fit well in their procedures, both for scenarios development and for near-
20  real time implementation. Practical applications can consist on determining new locations for emergency facilities, on increasing facility capacities, for traffic management planning or efficient and safer routing of emergency intervention vehicles." in the results and discussion section.
- We have added a preliminary qualitative assessment of uncertainties, in the (newer) section 3.3.
- We have modified figure 6 in order to move labels outside the map, to make it more visible, still pointing to the names
25  desired to highlight (which also appear in the text). We modified also Fig. 4.
- We kept the Romanian name version, because otherwise they will not be well understood by local stakeholders.

**Comparison: manuscript version 2 modifications to manuscript version 1**

[revised manuscript text omitted]

- the proper definition of the network, with detailed data regarding component characteristics and connectivity. One of the problems is still in most cases the lack of official data: in developed countries there can be available good and updated GIS databases, however in most other countries transportation network data (at least for roads or railways) is not well officially defined and/or shared with the general public, therefore alternative data sources need to be used, such as OpenStreetMap (open-source). Google Maps, Here Maps etc. There are currently many software solutions capable of network development (including AutoCAD Civil 3D, OpenRoads or ESurvey Road Network), but not so many with risk analysis capabilities; among them we mention popular solutions such as ArcGIS for Desktop with the Network Analyst extension, PTV Visum/Vissim, Maeviz/Eqvis or STREET;
- the determination of direct damage probability of individual components. For this, earthquake engineering analysis methods are mostly used, such as dynamic elastic and inelastic analysis using grids and numerical methods: finite element method, pushover or time-history analysis, response spectra etc. A good synthesis of these methods can be found in  Crowley et al. (2011) and Costa (2003);
- the need to define relevant performance indicators, reflecting time or cost differences between pre and post-disaster network behavior; many performance indicators for networks can be found in literature, some of the most common at system level being Driver Delay, Simple/Weighted Connectivity Loss (Pinto et al., 2012; Poljanšek et al., 2011), System Serviceability Index (Wang et al., 2010) or Serviceability Ratio (Adachi and Ellingwood, 2008).

In the  recent years, new technologies such as Internet of Things devices, Big Data, Remote Sensing, drones, low-cost sensors and Machine Learning started to be quickly adopted as they can provide practical solutions for transportation network data collection and analysis. It is expected that the impact of future natural hazards on transportation networks will be much better recorded (as shown by Voumard et al., 2018), allowing for a better validation of risk models and opportunities to create more representative methodologies for the analysis of network risk, also in near-real time.

In order to analyze systemic risk (and not only component risk), networks need to be evaluated from the perspective of direct damage implication on connectivity, traffic changes or new traffic flows created, leading to indirect damage. Recent studies have addressed these aspects (Koks et al., 2019; Vodak et al., 2015; Caiado et al., 2012; Bono and Gutierrez, 2011; Douglas et al., 2007; Franchin et al., 2006), going beyond the simple summarization of direct effects and eventually of reconstruction costs generated. These studies also highlight an important aspect to consider (Pitilakis and Kakderi, 2011): interactions between the components of the system (inter-interactions) and with components of other systems (intra-interactions).

After analyzing available methodologies and solutions in the field of study, we reached the conclusion that nowadays capabilities can be better exploited, enabling a more flexible but also standardized analysis of transportation network implications due to natural hazards, compared to previous works. We consider that many has been done theoretically and too little practically (at least at full city scale analysis), leaving also room for new technologies and that is what motivated us to create a new GIS solution sharable with the community and applicable world-wide. In this paper that provides a settled methodology after preliminary studies such as Toma-Danila (2018) or Toma-Danila et al. (2016), we will focus on:

- presenting a methodology for evaluating direct and indirect transportation network risk due to natural hazards, embedded in ArcGIS Desktop as an open-source toolbox called "Network-risk";
- demonstrating its capabilities for a representative case study:  Bucharest - one of the most under risk capitals in Europe due to the implications of earthquakes; results represent an important contribution to emergency management risk reduction planning.

**2 Methodology and implementation**

The generalized steps of the  methodology comprises of:
- defining a transportation network in a GIS;
- evaluating which segments could be affected by a natural hazard (directly or indirectly) - accounting also for the  probability of damage;
- generating random damaged network scenarios based on this probability;
- evaluating which are the implications, in terms of connectivity and serviceability losses and then socio-economic consequences.

This concept was previously defined in studies such as Hackl et al. (2018), Zanini et al. (2017), Vodak et al. (2015), Chang et al. (2012) or Argyroudis et al. (2005). However, the way each of the tasks are treated, linked and implemented in GIS is what we consider to be a progress toward standardization and usability in real situations (also in near-real time). The methodology presented in Fig. 1 allows among others the consideration of multiple transportation network types (road, railway, utilities etc., represented at local, regional or national level) and of different natural hazards. The methodology

155  can accommodate, for example, to the analysis of earthquake implications, where damage is widespread and building debris , traffic patterns and a good level of details for network definition are necessary to be considered. For landslides, the factors to be considered will change, since damage will be much more punctual and random simulations might not be so representative. For flooding, vulnerability analysis of networks such as road or railways will require knowledge on topography - not so representative for earthquake analysis. Still,

160 the methodology will accommodate all these hazard types and influences, as long as, for example, loss analysis will lead to the identification of possibly affected network segments. There is also flexibility in the way the risk analysis is oriented - toward emergency intervention, economic losses evaluation or urban planning.

Most of the input data (yellow boxes in Fig. 1) is required, also with GIS reference, with the exception that, depending on the analysis type, emergency intervention facilities or origin-destination (OD) pairs will not necessary be needed. In addition, an

165 analysis without typical traffic data can be performed, although it might be representative  just for night traffic conditions.

The process of building a consistent transportation network, from more or less complex datasets, is an essential part in every network analysis . To assist in this effort we created a guide, models and layer symbologies for properly converting and editing data partially manually, following also the ArcGIS Desktop Network Analyst extension

170  recommendations. An alternative solution can be to use the ArcGIS OSM editor (https://github.com/Esri/arcgis-osm-editor) for OpenStreetMap data (possible however to have limitations in expressing Z-elevation), GRASS GIS v.net or procedures such as Karduni et al. (2016). Eventually, the converted data is expected by Network-risk to be similar to the sample files provided on the Network-risk webpage. At the moment, the compulsory columns required in the analysis are "name", "oneway", "F_ZLEV", "T_ZLEV", "hierarchy", "maximum_speed", "FT_minutes" and

175 "TF_minutes" ". To these, further columns accounting for traffic, scenario travel times or lack of functionality due to natural hazard effects will  be added, depending on data availability and analysis type. In the process of defining the rules for the Network Dataset (ND), the Network-risk toolbox requires to add more evaluators beside the ArcGIS Network analysis extension defaults, with the most important being for obstructions (used in service area analysis in the impedance field to reveal inaccessible road areas) and other for different typical traffic scenarios or for economic costs.

180 Both pre and post-earthquake traffic data are highly important, since they show the typical functionality status of the network and the premises for new traffic congestions, immediately after an earthquake (with correlations also to road segments blocked by e.g. building debris or bridge collapse). Typical traffic data can be retrieved from local data sources (such as traffic management authorities) or from companies taking advantage of new device capabilities, such as Google Traffic, Here Traffic or Waze. This sources provide live (or statistical) data regarding traffic values

185 and reported incidents  although to integrate this data into our framework it is needed to convert this data into travel speed per road segment turn into barriers or restrictions GIS layers. Other solutions with near-real time analysis capabilities can be to use GPS data - from emergency vehicles , or the expertise of their drivers, especially for emergency management analysis.

The network layer represents the exposure; to evaluate the vulnerability of network segments to a specific natural hazard (or multi-hazard - the analysis can also take this dimension), it is required to associate failure probabilities . For individual structures (such as bridges, tunnels, pump facilities, electricity poles) or for buildings (including network buildings), vulnerability functions are commonly used to determine damage probability or even more: functionality loss or resilience functions such as closure time or recovery cost. Although it is recommended to use structure-specific (local) functions, considering particular properties of the structure and of the construction practices in the specific country/region, there are currently available fragility function libraries, collected and harmonized in projects such as Hazus , Syner-G or SERA, which can be associated, preliminary, to some of the assets in other region. In some cases, analyzing the probability of a building to collapse can be further linked to the probability of road blockage, due to debris for example (in case of earthquakes, there are equations in this purpose such as Santarelli et al., 2018; Zanini et al., 2017; Argyroudis et al., 2005 ; Moroux et al., 2004). Knowing where affected areas are also contributes to the evaluation of indirect risk, aiding for example to calculate the chance of people caught under debris to survive  using results of on-field studies such as  Hekimoglu et al. (2013), Coburn and Spence (2002) or Goncharov (1997).

After including references to the natural hazard, in the form of a maps with transferable values to  vulnerability functions, the result would be an evaluation of the direct possible damage  and as such a probability of network segment blockage. This can be used for generating random scenario simulations using the Monte Carlo approach (potential acknowledged by Burt and Graham, 1971), in order to test the behavior of the network in multiple probable situations. Assigning a probability of 100% for the failure of a network segment (indicating certain blockage  is useful for  worst-case scenarios or clear cases of vulnerability (for example: a highly vulnerable bridge which will certainly not withstand high acceleration values due to an earthquake or a road segment where rock falls happen even without a significant trigger). However, in most of the cases this probability will need to be smaller, allowing for random simulations to show multiple implication patterns. Also, post-disaster traffic can be considered independently for each simulation. Monte Carlo scenarios are usually supposed to come in large number ( hundreds or thousands of runs) and, depending on the size of the network, the amount of computational time is expected to be considerable. However,  the need for a vast number of Monte Carlo scenarios might not be really necessary. The existence of many viable detour routes in urban areas  or the small number of identified network segments expected to be highly damaged can determine the need of a smaller sample of Monte Carlo scenarios - that is why the stabilization of results must be traced.

For estimating post-event traffic patterns, it is needed to include assumptions providing travel speed modifications for road segments located close to affected areas, especially in urban agglomerations. Some hints for determining these patterns can be found in the work of  Zanini et al. (2017) or Chang et al. (2012). More complex approaches

relying on individual driver behavior simulations or decision patterns  as described in Asaithambi and Basheer (2017) or Munigety and Mathew (2016) can be implemented.

At the core of the network implication analysis there can be used different shortest path routing algorithms (by short not referring always to distance, but also to less risk), such as Dijkstra, A*, Johnson's Algorithm or Floyd-Warshall. In our implementation and case study we preferred the Dijkstra algorithm , which was used for computing the shortest distance (in real meters or costs) for various network configurations - pre and post event (for Service Area/Route/Closest Facility/OD Matrix analysis). This algorithm is widely used in systemic network analysis (Sniedovich, 2016), providing a good balance between precision and performance (Bast et al., 2016), being also chosen as preloaded algorithm in ArcGIS. Depending on user preferences, other algorithms can be applied – using for example an alternate approach relying on QGIS with pgRouting (https://pgrouting.org/) or A* in ArcGis. For Service Area analysis , used in the emergency intervention travel time evaluation, we recommended as analysis method using Detailed Polygon Generation, with results of prior analysis for identifying inaccessible network areas as barriers, since the results will better reflect small inaccessible areas.

The entire methodology is embedded in a toolbox called Network-risk, which currently runs under ArcGIS Desktop Advanced (10.1+ version) with the Network Analyst extension enabled, using ModelBuilder capabilities (Fig. 2). This toolbox takes advantage of the already available geo-processing and location-allocation algorithms and enables a standardized, non-hazard dependent and automated large-scale network risk analysis . In this direction, we acknowledge the previous works of Vodak et al. (2015), Pinto et al. (2012) or Sevtsuk and Mekonnen (2012), which we consider however not fully usable especially in the more recent context. Having the methodology implemented in ArcGIS offers extended analysis support, through cartographic, spatial analyst modules, available basemaps, plug-ins such as ArcCASPER (Shahabi and Wilson, 2014) for computing evacuation routes and others. We chose to split Network-risk in multiple separate modules (such as for network creation, Monte Carlo scenario creation, disrupted network building, service area analysis or aggregation of results into a final index), making it easy to identify errors at different steps. The toolbox is available for download at www.infp.ro/network-risk and is free to use and customize.

Considering the steps described in Fig. 2, ArcGIS Network Analyst capabilities and the results which are later shown by our case study, Network-risk toolbox is capable to answer important questions for emergency management, city planning, commercial, insurance, industrial or real-estate agents and many others, such as:

-
- Which areas could become inaccessible after a natural disaster? Which are the vital access routes in case of a disaster?
- Are there viable detour routes?

- Which is the socio-economic impact (in terms of human or financial losses) in case of a natural disaster, correlated also with emergency management capabilities?
-
- How would new network segments, hospitals  fire stations or other facilities contribute to reducing the risk? Where should they be placed?

**3 Bucharest road network case study, considering seismic hazard**

**3.1 Case study area description**

For testing the methodology, we selected Bucharest - one of Europe's endangered capitals due to high seismic risk  (Toma-Danila and Armas, 2017; Pavel, 2016). The city affected by strong earthquakes in the Vrancea seismic area (such as the ones on November 10th, 1940, Mw 7.7 at 150 km depth and on March 4th, 1977, Mw 7.4 at 94 km depth) and is currently still poorly prepared (Pavel, 2016) for a next major event which will most certainly happen anytime in the next 100 years. Compared to 1977 (when 1578 people died in Romania, from which 90% in Bucharest), the city now faces an additional challenge, beside the high vulnerability of the building stock: the vulnerability due to road network and urban traffic. In a city with over 2 million inhabitants  there are 1.2 million registered vehicles (NIS, 2018). To this number can also be added the contribution of transit vehicles not adequately serviced by an external ring road (Fig. 3c) or vehicles of numerous commuting persons from nearby counties or students. In the absence of efficient urban development and mobility measures, in combination with mentality issues (the self-requirement to own and use a car), the city faces regular traffic jams, being ranked as Europe's number 1 capital (and 5th in the world in 2017/11th in 2018) when it comes to typical congestion level (TomTom, 2018; typical traffic examples in Fig. 3d, 3e and 3f). Beside traffic, Bucharest's road network maintenance and serviceability status is precarious, with many dysfunctions related to the quality of embankment, bridges, over or underpasses (Fig. 3a), poor repairing works, limitations in the full utilization of road's length due to illegal (and unsanctioned) parking in many cases (Fig. 3b) or constantly exceeded deadlines for repair or new road works. Another important aspect is that many buildings, not solely in the city center, are highly vulnerable to earthquakes (Toma-Danila et al., 2017). More than 31430 residential buildings were constructed prior to 1946 (294 having more than 4 storeys - a vulnerable category due to long fundamental periods of intermediate-depth Vrancea earthquakes), according to the 2011 National Population and Housing Census. In addition, 26349 residential buildings (237 having more than 4 storeys) were constructed between 1946 and 1960 , in a period with no compulsory seismic design code, enduring at least one major earthquake with limited evaluation and seismic retrofitting afterward (Georgescu and Pomonis, 2018). One should realize that if only 1% of them will completely or partially collapse, it could clearly lead to many deaths and injuries, difficult to manage considering hospital capacity and equipment, as the recent Colectiv Club fire disaster proved (Marica, 2017), but also due to severe road blockages. In 1977, central boulevards (such as Magheru)

were closed for at least 3 days after the March 4th earthquake, still the typical traffic was not severely affected due to low traffic values and the wide use of public transport in those days. We aim to how that nowadays, such a measure would have much more adverse implications. Considering also the nowadays much wider expected damage scale (Armas et al., 2016; Pavel and Vacareanu, 2016), emergency interventions will have to be provided from multiple locations (inside and outside the city) and usual traffic patterns (not to mention the ones right after a major earthquake, depending also on the time of occurrence) will clearly act against proper reaction. All these problems make Bucharest a highly representative testbed for the methodology proposed in this article.

Preliminary analysis of the associated seismic risk of the Bucharest road network was performed in the recent years, using slightly different approaches (Toma-Danila, 2018; Ianos et al., 2017), however not so flexible or at full city scale, concentrating only on the city center. Our goal for the analysis is to play an important role in the mitigation of seismic risk in Bucharest, being the first analysis for entire Bucharest.

**3.2 Data and methods considered for Bucharest**

The starting point for the analysis was the development of a road network GIS database, respecting connectivity and elevation rules. Currently, an official database of such kind is not available for Bucharest. That is why we used data from OpenStreetMap (OSM), which is one of the most successful crowdsourcing project aiming to create a geospatial database of the whole world, with relatively up-to-date data for Romania thanks also to the involvement of many local volunteers (https://forum.openstreetmap.org/index.php), with good applicability in vehicle routing (Graser et al., 2014). OSM road vector data was downloaded using the Geofabrik GIS Data Portal (http://download.geofabrik.de), requiring additional processing in ArcGIS Desktop's ArcMap (Network-risk toolbox template and guidelines are provided), in order to convert it in the ArcGIS network format, accounting for connectivity, hierarchy, travel direction (From-To - FT and To-From - TF), Z-elevation (creating distinctions between roads at ground level, bridges or underpasses) and travel time. For Bucharest - up to the external ring road and its connections to city center, the final number of individual road segments resulted (everything represented in Fig. 3) was 50412. We used data from September 2016; since then up to December 2019 no major road network modifications happened in Bucharest (the main exception being the extension of A3 up to north-eastern Bucharest, but with no major influence on our analysis). When analyzing statistics (especially road length) it is important to account for road segments difference of drawing roads per lane or as a whole in OSM - otherwise the real number of kilometers will in some cases be doubled. That is why we prefer not to present statistical road length graphs.~~In the process of defining the rules for the Network Dataset (ND), the Network-risk toolbox requires to add more evaluators beside the ArcGIS Network analysis extension defaults, the most important being for the obstructions (if a road segment is affected, according to the scenario, the service area analysis using as impedance this evaluator will reveal inaccessible road areas), and other being for different typical traffic scenarios or for economic costs.~~

For determining (also in a Monte Carlo manner) which road segments can be affected by earthquakes  we used the procedures described in Table 1.  In total were determined, totaling 1.41% of the total number of road segments in Bucharest:

- 1324 segments with variable length which can become affected by debris (partially shown in Fig. 4, just 32,6% however
325 with a damage probability  > 50%);
- 985 which can become affected by bridge collapse.

After performing  20 Monte Carlo simulations (each with an average runtime of 12 minutes on a normal desktop computer  – from simulation to service area results), we considered results stable enough to reflect the damage patterns for the rather extended road network of Bucharest.
330

 and stopped our simulations, which are not time intensive but still difficult to summarize automatically. In order to account for traffic -  a major issue for Bucharest, we followed the patterns shown by typical  Google Traffic, for various representative scenarios:

335 - Monday 2:00 AM - no traffic;
- Monday 8:00 AM - morning traffic;

[revised manuscript text omitted]

450  routes or best facility in terms of safe to reach proximity. Useful maps or routing services which could become available in near-real time can be obtained by combining the fastest routes for OD pairs, for a given scenario, showing also which roads are vital in an emergency situation (need to remain functional since they are critical, providing the quickest access time in the origin).
455

The seismic risk due to road network dysfunctionalities can be expressed not just by considering the impact of road blockage and traffic on emergency intervention, leading to time limitations in reaching patients. When roads are closed, connectivity throughout the city can be lost for days, weeks or years, with a high impact on economy - due to delays in stock supplies and production, greater costs for carburant or loss of clients. The createdOur network dataset can also be used also to monitor which are the differences between pre and post-earthquake travel times, for representative OD pairs. For this case study we selected 8 pairs in relevant to cardinal points, some with links to the city center and some aimed to show if in case of an earthquake the initially preferred route throughout the city is going to change in favor of the external ring road.

Uncertainties and limitations are an important aspect to account for. As a preliminary evaluation we provide the following qualitative uncertainty evaluation, regarding:

- the road network dataset accuracy - small source of uncertainties;
- the limited dataset regarding buildings which could collapse during an earthquake – moderate source of uncertainties;
- limitation in evaluating and validating the travel times for emergency intervention vehicles (as recently the allowance of using tramway separated tracks lead to improved intervention times) – moderate source of uncertainties.
- typical traffic scenarios considered – small source of uncertainties;
- post-earthquake traffic patterns – high source of uncertainties.

**3.3 Results are**

The figures presented in this subchapter summarize our main findings and are obtained for the multiple Monte Carlo and worst-case scenarios run with Network-risk toolbox. Results are foreseen to contribute to:

- operational procedures of the Inspectorates for Emergency Situations (such as the National Concept for Post-Earthquake Intervention - implementation discussion on-going);
- risk-reduction strategies elaborated at national and local level;
- the new planning of new emergency hospitals in Bucharest;
- the identification of easy to access locations for emergency containers.

Figure 5a and 5b reflect differences between worst-case scenario (all roads and bridges with a probability of damage affected) and results from Monte Carlo simulations. As such, Fig. 5a presents, for some areas, slightly more increased intervention time values. Figure 5c shows service area intervals when considering only emergency hospitals in category I of importance. It can be seen that their distribution is generally satisfactory, however there is an area with significantly greater intervention times, reflected also by Fig. 5a and 5b, in the south-west area of Bucharest (Rahova and Ferentari neighborhoods) - an area known also for its socio-economic vulnerability (Armas et al., 2016), also with no major hospital in adjacency. Due to the significant damage expected in the central area, intervention times are expected to be considerable (given also the traffic values for the considered scenario). The impact of a central hospital such as Coltea is reflected in the partial decrease of ambulance intervention times for city center. However, in the post-earthquake chaos, especially if the earthquake will strike at rush hour, traffic jams are going to pose a considerable threat to road accessibility; our study reveals some of these effects (Fig. 5-9) and

that some areas could be much easier accessed by ambulances from non-central locations. Bridge dysfunctionalities do not seem to pose great influences (when comparing also with no damaged bridge scenarios), since in general there are many nearby alternatives. Basarab Overpass (north-west to the center - labeled in Fig. 3) is the only one who, if inaccessible, could lead to considerable increase of intervention times. Figure 5d is, although difficult to comprehend at first sight, important since it provides a visual check upon the correlations between minimum intervention times and the number of hospitals who provide this time; if an area is colored towards green and is also hatched, this means that the area is close to multiple emergency hospitals, having a lower vulnerability in case of medical emergencies. Data behind this type of maps adds an additional understanding to the overall accessibility analysis, being however more demanding in their creation (requiring service area analysis per facility and counting of number of overlapping polygons with a certain value).

Figure 6 shows service area results for fire stations; the distribution of fire stations is more symmetrical in Bucharest then the distribution of hospitals, also with a unit in the city center ("Mihai Voda" fire department), behind the Bucharest City Hall building. For the chosen scenario (Monday 8AM typical traffic), the influence of this distribution can be seen south of Piata Unirii (Fig. 6b zoom map), were also boulevards are not expected to be blocked by debris, but north - toward Piata Universitatii and Piata Romana, post-earthquake congestion and road segment blockages are expected to significantly increase the travel times. To help in the effort of reducing the intervention times in the central area, the "Victoria Palace" fire department (devoted to the Government's building) could contribute, however we did not find appropriate at the moment to consider it in the analysis, until learning more about their attributions.

In order to facilitate the understanding of results, also from the point of view of non-experts, we further show the results of the aggregation methodology used for creating a final index of vulnerable road accessibility for Bucharest. Figure 7 - the first map of this kind for the entire territory of Bucharest, reflects some of the expected features: a high vulnerability of accessibility in central area of the city, due to vulnerable buildings and difficult to reach (in case of an earthquake) hospitals (especially in Category I of importance) and fire stations. Also, the figure shows other areas more difficult to reach by all types of emergency vehicles right after an earthquake: western Bucharest (Militari neighborhood) or south-western and south-eastern Bucharest. Areas with good accessibility appear to be in the inner green belt north to the inner ring road – where there are hospitals and fire stations nearby and no disruptive traffic (although quite intense during rush-hours) and disrupted road segments.

Another important result of the analysis, proving the Network-risk capabilities, is presented in Fig. 8. As expected, inaccessible areas are mostly in the city center (streets such as Blănari, Lipscani, Şelari, Smârdan, Sf. Dumitru, Franceză, Tonita, Eforiei or Biserica Doamnei), where many buildings are expected to block roads and detour routes to the locations. Other blocked road segments, with lower probability, could be on streets such as Bărăției, Pătrașcu Vodă, Vasile Lascăr, Poiana Narciselor, Dr. Vasile Sion, Ion Brezoianu, Tudor Arghezi, Batiștei, Jules Michelet etc. Due to the algorithm for Service Area computation, some areas between roads are colored as being blocked (as in Cismigiu Central Park for example), however this is a method limitation and can be eliminated through clipping.

Fig. 9 is the result of Closest Facility Analysis, showing the safest and fastest routes (and the density of these routes) between buildings in seismic risk class I and emergency hospitals and which hospitals would be the preferred facility for a certain

building, based on adjacency (no medical capabilities are considered) - setting premises for a better preparedness of hospitals expected to have a high patient demand (medical supplies, hospital beds, doctors etc.). Figure 9a highlights, for the specific scenario, 3 routes in high demand: from city center toward east, west and north-west. Figure 9b shows that Coltea Hospital is not although in the city center, not the best option for many vulnerable buildings.

Table 5 and  Fig. 10 show results for representative OD pairs to the economic transit routes – any other OD pairs can be introduced. For the 2 AM traffic scenario, differences are not significant, as post-earthquake traffic is not expected to be a significant problem, however for the 8 AM and 6 PM scenarios - especially for routes which need to reach the city center (Piata Universitatii for example), there are clear values showing a mean travel time increase from 110-120% to 300-432%, for the Centura (external ring road) - Otopeni -> Piata Universitatii route.

**4 Conclusions**

In this paper we presented a new methodology for evaluating  direct and indirect implications of natural hazards on transportation network. This methodology was designed to be generally applicable and adaptive to various types of hazards, networks or available vulnerability and exposure data. Starting from structural evaluation, the analysis focuses on systemic or functional assessment, expressing furthermore the risk inflicted mainly by connectivity loss. After determining hazard, exposure and vulnerability  factors - leading to the definition of the network and the identification of segments which can become unusable (and the probability of this to happen),  Monte Carlo simulations can be performed. This enables the creation of multiple scenarios evaluated individually in terms of generated risk (for emergency intervention or socio-economic aspects) and aggregated into final risk indexes. There are also capabilities of accounting for pre and post disaster traffic and for emergency facilities capacity or equipment. In order to facilitate the use of the methodology we ~~also integrated it into an open toolbox (collection of models) - free to download and customize, entitled Network-risk (available on www.infp.ro/network-risk). This toolbox is for now dependent on the geoprocessing algorithms implemented in the widely used commercial software ArcGIS Desktop Advanced, with Network Analyst extension. In the near future we will try to integrate Network-risk also in non-commercial GIS software such as QGIS, who still require at the moment more development toward advanced network analysis. Network-risk toolbox is under continuous development and in future versions more features will be available, so please check regularly the website~~integrated it into an open toolbox (collection of models) entitled Network-risk, which is free to download and customize.

To prove its capabilities, Network-risk was tested on the entire road network of Bucharest, Romania, one of Europe's most endangered capitals due to earthquakes, considering the high seismic hazard values generated by intermediate-depth Vrancea earthquakes, the vulnerable building stock (349 high or moderate rise buildings are categorized in the seismic risk class I in January 2016, representing just the tip of the vulnerability "iceberg") but also major traffic congestion patterns. One of the most difficult parts in the analysis was the proper input data collection. As we showed, this can be achieved (at least for a

555 preliminary form)  in a satisfactory form, by using OpenStreetMap data along with a Network-risk module designed to arrange (partially automatically) the network data into ArcGIS network format. Digitized traffic areas based on Google Traffic layers or empirical formulas, literature fragility functions and expert judgement for determining road segment failure probabilities also contribute to the input. Our analysis focused both on the evaluation of emergency intervention times (for emergency hospitals and fire stations) and on the evaluation of economic implications for

560 representative commercial routes (time delays in post-earthquake conditions).

Results show that the city center would be significantly vulnerable not just because of collapsing buildings but also due to the difficulty to reach these sites by ambulances and firefighters; although there are facilities nearby, such as the Coltea Hospital (however not of category of importance I) and the "Mihai Voda" fire department, these do not provide safe routes to all potentially affected buildings, due to road blockages and traffic jams, considering especially the Monday 8AM and 6PM typical

565 traffic scenarios. Aggregated results in Fig. 7 and 8 show that also for the western, south-western and south-eastern parts of Bucharest overall intervention times can be significant.  – valid supposition confirmed verbally by members in the emergency intervention forces.

**4.1 Discussion**

570 Stakeholders such as emergency situations managers provided us important feedback, acknowledging that the final products can fit well in their procedures, both for scenarios development (prevention) and for near-real time implementation (reaction). Practical applications can consist on determining new locations for emergency facilities, on increasing facility capacities , for traffic management planning.

575 safer routing of emergency intervention vehicles. As a comment for future methodology users, we want to mention that, when calculating service areas, it is very useful to account for the dependency to a single facility to provide the minimum intervention time and we will aid a module in Network-risk to provide a performance indicator in this purpose.

In our opinion, the service area analysis for Bucharest shows the necessity of an emergency hospital in the south-western part of Bucharest - an area also known for its high socio-economic vulnerability

580 . For the city center, a strategy in case of an earthquake has to be elaborated and put into place, referring to  measures to facilitate/restrict the access in the area in case of natural disasters, traffic redirection and design of safe road access corridors. As highlighted, the vulnerability of routes connecting the city center, especially with north or south destination, can be significant, with travel time increase greater than 150% in typical scenario conditions.

585

As Network-risk is for now dependent on the commercial software ArcGIS Desktop Advanced, with Network Analyst extension, we will try in the near future to integrate its methodology also in non-commercial GIS software such as QGIS. However, this still require at the moment more development toward advanced network analysis. The current Network-risk toolbox is under continuous development and in future versions more features will be available, so please check regularly the website. We also aim to test it more consistently, with analyzes at regional/national scale (using also rapid seismic loss estimations generated by the Seisdaro System of INFP, presented by Toma-Danila et al., 2018), for multiple hazard scenarios and also for more detailed vulnerability datasets comprising also on the social behavior and interaction of people with transportation networks.

We hope that this article will provide researchers important practical guidelines on how to analyze the risks of transportation networks affected by natural hazards and a useful tool to be applied in other parts of the world and stakeholders an example of useful results which they could benefit from, in their efforts to better understand and mitigate risks.

**Code and data availability**

[revised manuscript text omitted]

Graser, A., Straub, M., and Dragaschnig, M.: Is OSM Good Enough for Vehicle Routing? A Study Comparing Street Networks in Vienna, in: Progress in Location-Based Services, edited by: Gartner, G., and Huang, H., Springer, Cham, 3-17, 2014.

Goncharov, S.F.: Medical consequences of earthquake disasters in Russia; Earthquake and People's Health, in: Proceedings of the WHO Symposium, Kobe, Japan, 1997.

Gu, D.: Exposure and vulnerability to natural disasters for world's cities; United Nations Department of Economic and Social Affairs, Population Division, Technical Paper No. 2019/4, 2019.

Hackl, J., Lam, J. C., Heitzler, M., Adey, B. T., and Hurni, L.: Estimating network related risks: A methodology and an application in the transport sector, Nat. Hazards Earth Sys., 18, 2273–2293, 2018.

Hekimoglu, Y., Melez, I.E., Canturk, G., Erkol, Z., Canturk, N., Dizdar, M.G., Melez, D.O., Guler, O.S.: Evaluation of the deaths secondary to entrapment under the debris in the Van earthquake, Egyptian J Forensic Sci, 3(2), 44-47, 2013.

Ianos, I., Merciu, G.L., Merciu, C., and Pomeroy, G.: Mapping Accessibility in the Historic Urban Center of Bucharest for Earthquake Hazard Response, Nat. Hazards Earth Sys., Discussion Paper, doi: 10.5194/nhess-2017-13, 2017.

Jenelius, E., Mattsson, L.G.: Road network vulnerability analysis: Conceptualization, implementation and application, Comput Environ Urban, 49, 136–147, 2015.

Karduni, A., Kermanshah, A., and Derrible, S.: A protocol to convert spatial polyline data to network formats and applications to world urban road networks, Sci Data, 3, 160046, 2016.

Kiremidjian, A., Moore, J., Fan, Y.Y., Yazlali, O., Basoz, N., and Wiliams, M.: Seismic Risk Assessment of Transportation Network Systems, J. Earthq. Eng., 11(3), 371-382, 2007.

Koks, E.E., Rozenberg, J., Zorn, C., Tariverdi, M., Vousdoukas, M., Fraser, S.A., Hall, J.W., and Hallegatte, S.: A global multi-hazard risk analysis of road and railway infrastructure assets. Nat Commun, 10, 2677, 2019.

Lerner, E.B., and Moscati, R.M.: The Golden Hour: Scientific Fact or Medical "Urban Legend?, Acad. Emerg. Med., 8(7), 758–760, 2001.

Marica, I.: Report on 2015 Bucharest club tragedy unveils healthcare system problems, available at https://www.romania-insider.com/report-2015-colectiv-club-tragedy, last access: 1 December 2019, 2017.

Marmureanu, G., Cioflan, C.O., and Marmureanu, A.: Researches on local seismic hazard (Microzonation) for metropolitan Bucharest area (in Romanian), Tehnopress, Iasi, Romania, 2010.

Miller, M.K.: Seismic Risk Assessment of Complex Transportation Networks, 2014.

Moroux, P., Bertrand, E., Bour, M., LeBrun, B., Depinois, S., Masure, P., and Risk-UE team: The European Risk-UE Project: An advanced approach to earthquake risk scenarios, in: Proceedings of the 13th WCEE, Vancouver, Canada, 2004.

Munigety, C.R., and Mathew, T.V.: Towards Behavioral Modeling of Drivers in Mixed Traffic Conditions, Transportation in Developing Economies, 2, article nr. 6, 2016.

Neagu, C., Arion, C., Aldea, A., Calarasu, E.A., Vacareanu, R., and Pavel, F.: Ground Types for Seismic Design in Romania. in: Seismic Hazard and Risk Assessment, edited by: Vacareanu, R., and Ionescu, C., Springer Natural Hazards, Springer, Cham, 157-172, 2018.

NIS (National Institute of Statistics): TEMPO-Online database, available at http://statistici.insse.ro:8077/tempo-online/, last access: 1 December 2019, 2018.

Pavel, F.: Next Future Large Earthquake in Romania: A Disaster Waiting to Happen?, Seismol. Res. Lett., 88(1), 1-3, 2016.

Pavel, F., and Vacareanu, R.: Scenario-based earthquake risk assessment for Bucharest, Romania, Int. J. Disast. Risk Re., 20, 138-144, 2016.

Pesaresi, M., Ehrlich, D., Kemper, T., Siragusa, A., Florczyk, A., Freire, S., and Corban, C.: Atlas of the Human Planet 2017: Global Exposure to Natural Hazards, Publications Office of the European Union, doi: 10.2760/19837, 2017.

Pinto, P.E., Cavalieri, F., Franchin, P., and Lupoi, A.: Systemic vulnerability and loss for transportation systems, D5.5 of the SYNER-G Project, 2012.

Pitilakis, K.D., and Kakderi, K.G.: Seismic risk assessment and management of lifelines, utilities and infrastructures, in: Proceedings of the 5th ICEGE, Santiago, Chile, 2011.

Poljanšek, K., Bono, F. and Gutiérrez, E.: Seismic risk assessment of interdependent critical infrastructure systems: The case of European gas and electricity networks, Earthq. Eng. Struct. D., 41(1), doi: 10.1002/eqe.1118, 2011.

Santarelli, S., Bernardini, G., and Quagliarini, E.: Earthquake building debris estimation in historic city centres: From real world data to experimental-based criteria, Int J Disast Risk Re, 31, 281-291, 2018.

Sartori, M.: Seismic protection of the Basarab overpass in Bucharest, in: Proceedings of the 15th WCEE, Lisboa, Portugal, 2012.

Sevtsuk, A., and Mekonnen, M.: Urban network analysis. A new toolbox for ArcGIS, Rev Int Géomatique, 22, 2, 287-305, 2012.

[revised manuscript text omitted]